# Fan cells in lateral entorhinal cortex directly influence medial entorhinal cortex through synaptic connections in layer 1

Brianna Vandrey[1]*, Jack Armstrong[1], Christina M Brown[1], Derek LF Garden[1], Matthew F Nolan[1,2,3]*

[1]Centre for Discovery Brain Sciences, University of Edinburgh, Edinburgh, United Kingdom; [2]Simons Initiative for the Developing Brain, University of Edinburgh, Edinburgh, United Kingdom; [3]Centre for Statistics, University of Edinburgh, Edinburgh, United Kingdom

**Abstract** Standard models for spatial and episodic memory suggest that the lateral entorhinal cortex (LEC) and medial entorhinal cortex (MEC) send parallel independent inputs to the hippocampus, each carrying different types of information. Here, we evaluate the possibility that information is integrated between divisions of the entorhinal cortex prior to reaching the hippocampus. We demonstrate that, in mice, fan cells in layer 2 (L2) of LEC that receive neocortical inputs, and that project to the hippocampal dentate gyrus, also send axon collaterals to layer 1 (L1) of the MEC. Activation of inputs from fan cells evokes monosynaptic glutamatergic excitation of stellate and pyramidal cells in L2 of the MEC, typically followed by inhibition that contains fast and slow components mediated by $GABA_A$ and $GABA_B$ receptors, respectively. Inputs from fan cells also directly activate interneurons in L1 and L2 of MEC, with synaptic connections from L1 interneurons accounting for slow feedforward inhibition of L2 principal cell populations. The relative strength of excitation and inhibition following fan cell activation differs substantially between neurons and is largely independent of anatomical location. Our results demonstrate that the LEC, in addition to directly influencing the hippocampus, can activate or inhibit major hippocampal inputs arising from the MEC. Thus, local circuits in the superficial MEC may combine spatial information with sensory and higher order signals from the LEC, providing a substrate for integration of 'what' and 'where' components of episodic memories.

**\*For correspondence:**
brianna.vandrey@ed.ac.uk (BV);
mattnolan@ed.ac.uk (MFN)

**Competing interest:** The authors declare that no competing interests exist.

## Editor's evaluation

This is an important manuscript that convincingly reveals a novel pathway by which the lateral entorhinal cortex directly projects to the medial entorhinal cortex. This work thus revises the traditional models that envision lateral and medial entorhinal cortex as providing independent inputs to the hippocampus. Instead, the work points to these cortical regions as participating in the combination of spatial information with sensory and other high-order signals even before routing information to the hippocampus for memory formation.

## Introduction

The anatomical organisation of the hippocampus and its associated structures imposes fundamental constraints on mechanisms for spatial cognition and memory. According to standard models,

information about spatio-temporal context is processed through medial entorhinal cortex (MEC), while information about objects and events is processed in parallel through the lateral entorhinal cortex (LEC), with both streams being integrated in the hippocampus (*Behrens et al., 2018*; *Burwell et al., 2004*; *Eichenbaum et al., 2012*; *Knierim et al., 2014*; *Nilssen et al., 2019*; *Whittington et al., 2020*). Consistent with this framework, neurons in the MEC represent spatial variables including location (*Fyhn et al., 2004*; *Hafting et al., 2005*; *Hardcastle et al., 2017*; *Høydal et al., 2019*), head direction (*Sargolini et al., 2006*), and running speed (*Kropff et al., 2015*), whereas neurons in the LEC encode information about local features of the environment (*Deshmukh et al., 2012*; *Deshmukh and Knierim, 2011*; *Igarashi et al., 2014*; *Keene et al., 2016*; *Kuruvilla and Ainge, 2017*; *Lee et al., 2021*; *Tsao et al., 2018*; *Tsao et al., 2013*; *Wan et al., 1999*; *Xiang and Brown, 1999*; *Young et al., 1997*; *Zhu and Brown, 1995*). However, recent experiments suggest that spatial and non-spatial signals may be combined in the entorhinal cortex, upstream from the hippocampus. Spatial cells in the MEC, including grid cells, can encode information about object identity (*Keene et al., 2016*) and the superficial layers of the MEC contain neurons that encode vector relationships between objects and an animal's position (*Andersson et al., 2021*; *Høydal et al., 2019*). These findings raise the question of whether entorhinal circuits upstream from the hippocampus are organised to support integration of spatial and non-spatial signals.

Projections from layer 2 (L2) of the MEC and LEC converge on common postsynaptic targets within the dentate gyrus and CA3 regions of the hippocampus (*Hainmueller and Bartos, 2020*; *Steward, 1976*). These projections primarily originate from reelin expressing neurons, which in MEC and LEC are referred to respectively as stellate cells (L2 SCs) and fan cells on the basis of their distinct dendritic morphology (*Alonso and Llinás, 1989*; *Dolorfo and Amaral, 1998*; *Klink and Alonso, 1997*; *Leitner et al., 2016*; *Nilssen et al., 2018*; *Tahvildari and Alonso, 2005*; *Vandrey et al., 2020*; *Varga et al., 2010*). An additional projection to the CA1 region of the hippocampus arises from calbindin expressing pyramidal cells (L2 PCs) (*Kitamura et al., 2014*; *Ohara et al., 2019*; *Sürmeli et al., 2016*), which in MEC form clusters that intermingle with L2 SCs (*Kitamura et al., 2014*; *Ray et al., 2014*; *Varga et al., 2010*), and in LEC form a separate sub-layer located deep to the fan cell layer (*Leitner et al., 2016*; *Vandrey et al., 2020*). Stellate cells in the MEC play important roles in contextual and spatial memory and are a major population of grid cells (*Gu et al., 2018*; *Qin et al., 2018*; *Rowland et al., 2018*; *Tennant et al., 2018*), while pyramidal cells are important for temporal associative memory (*Kitamura et al., 2015*; *Kitamura et al., 2014*). In contrast, fan cells in LEC have been shown to contribute to the processing (*Leitner et al., 2016*) and memory encoding of olfactory information (*Lee et al., 2021*), learning of object locations (*Fernández-Ruiz et al., 2021*), and processing of object-place-context associations (*Vandrey et al., 2020*). Neurons in the superficial layers of the LEC also generate representations of objects and object locations (*Deshmukh et al., 2012*; *Deshmukh and Knierim, 2011*; *Tsao et al., 2018*), although these are yet to be mapped onto a specific type of neuron. Within both LEC and MEC, layer 2 principal neurons interact through nearby fast spiking inhibitory interneurons (*Couey et al., 2013*; *Nilssen et al., 2018*; *Pastoll et al., 2013*), such that inputs that target either principal cells or interneurons may in principle shape the entorhinal input to the hippocampus.

While signals from MEC and LEC are generally considered to be integrated in the hippocampus, classic anatomical observations suggest that information could in principle be transferred directly between the MEC and LEC (*Köhler, 1988*). Specifically, injection of anterograde tracers into the superficial LEC of the rat labelled axons that pass through the MEC. However, it is unclear which neurons in the LEC this projection originates from, whether the projection makes functional synaptic connections within the MEC, and if so, what the identity of the postsynaptic targets are. These features of the circuit architecture impose fundamental constraints on its possible functions. For example, if this pathway contributes to processing of information before it is delivered to the hippocampus, then it should connect neurons in LEC with hippocampal projecting neurons in the MEC. In this case it will also be important to establish how the pathway interacts with local excitatory-inhibitory microcircuits which appear critical for computations in superficial layers of MEC (*Couey et al., 2013*; *Miao et al., 2017*; *Pastoll et al., 2013*; *Shipston-Sharman et al., 2016*). If the pathway drives postsynaptic activation of principal cells in the MEC then it could promote activation of convergent inputs to the downstream hippocampus, whereas if the pathway inhibits principal cells then it would oppose the influence of the MEC on the hippocampus.

It is of particular interest that axons from LEC appear to primarily pass through layer 1 ( L1) of MEC (*Köhler, 1988*), as this layer contains dendrites from L2 SCs and pyramidal cells with cell bodies in deeper layers (*Canto and Witter, 2012*; *Hamam et al., 2000*; *Klink and Alonso, 1997*; *Sürmeli et al., 2016*; *van Haeften et al., 2003*). A similar circuit arrangement has functional implications in neocortex, where L1 contains sparse populations of GABAergic inhibitory interneurons with distinct electrophysiological and morphological profiles (*Jiang et al., 2015*; *Schuman et al., 2019*). Neocortical L1 is targeted by various long-range subcortical and cortical inputs and is a locus for integration of external information about the environment and internally generated signals (*Schuman et al., 2021*). Activation of neurons in L1 of neocortex enables long range inputs to exert powerful inhibitory effects on principal neurons in deeper layers (*Jiang et al., 2015*; *Oláh et al., 2009*; *Schuman et al., 2019*; *Wozny and Williams, 2011*). In contrast, very little is known about the external inputs to L1 neurons in the MEC or how they influence activity of neurons in other layers.

Here, we show that projections from LEC to MEC are mediated by axon collaterals of LEC fan cells. These projections are localised to L1 of MEC, and arise from neurons in LEC that receive synaptic input from piriform and prefrontal cortices. Activation of fan cell inputs to MEC drives excitatory postsynaptic potentials, in L2 SCs and L2 PCs, that are mediated by direct glutamatergic input, and feedforward inhibition that arises from local interneurons in L1 and L2. Inhibitory components of responses to input from fan cells are particularly prominent in L2 SCs, and are composed of fast and slow components mediated by $GABA_A$ receptors and $GABA_B$ receptors, respectively. The relative balance of excitation and inhibition is heterogeneous between neurons and appears largely independent of anatomical location. By delineating circuit mechanisms through which MEC can integrate inputs from the LEC, our results identify a pathway through which sensory and higher order information from neocortex may contribute to spatial and object coding in the superficial MEC.

## Results

### Fan cells in LEC project to superficial MEC

As a first step toward determining the identity of neurons in the LEC that project to the MEC, we injected a retrograde AAV encoding mCherry (AAV-Retro-mCherry) into the MEC of wild-type mice (*Figure 1A*, n=3 mice). Labelled neurons in the LEC were primarily localised to the most superficial sublayer of layer 2 (L2a, 21.6 ± 1.6% of neurons in L2a), with sparse labelling of neurons in layer 5a, suggesting that the majority of axonal projections from LEC to MEC arise from fan cells in L2a. To test this directly, we injected adeno-associated virus (AAV) mediating Cre-dependent expression of GFP (AAV-FLEX-GFP) into the LEC of *Sim1*[Cre] mice (n=4), which we found previously to give specific genetic access to fan cells in the LEC (*Vandrey et al., 2020*). This strategy led to GFP labelling of fan cell bodies in L2a of the LEC, and labelling of fan cell axons in L1 of the MEC in addition to their established termination zone in the outer molecular layer of the dentate gyrus (*Figure 1B*). Thus, projections from the LEC to the MEC originate from fan cells.

Denser labelling of fan cells axons in a medial strip of MEC bordering parasubiculum (*Figure 1B*) suggests that these projections may differentially target neurons across the mediolateral axis of MEC. Because axonal labelling with GFP may not reflect the presence of synaptic terminals, we adopted a second approach to labelling fan cell axon terminals using synaptophysin-mRuby. We injected AAV mediating Cre-dependent expression of synaptophysin-mRuby and GFP (AAV-FLEX-GFP-2A-Synaptophysin-mRuby) into the superficial LEC of a further cohort of *Sim1*[Cre] mice (n=3). Quantification of the density of synaptic terminals confirmed a striking enrichment in L1 compared to all other layers, low levels of labelling in L3 and L5a, and slightly elevated labelling in L2 and L5b (*Figure 1C–D*). Comparison of labelling within L1 again revealed a greater density of puncta in the region of the MEC adjacent to the parasubiculum (*Figure 1D–E*).

The projections to the MEC that we identify here could in principle arise from a distinct subpopulation of fan cells, or they could reflect collaterals of fan cell axons projecting to the dentate gyrus. To distinguish these possibilities, we targeted GFP expression to hippocampal-projecting fan cells by injecting retrograde AAV encoding Cre (AAV-Retro-Cre, *Gradinaru et al., 2010*) into the dentate gyrus and AAV encoding Cre-dependent GFP into the superficial LEC of wild-type mice (n=3) (*Figure 2A, C*). This approach revealed labelling of axons in the dentate gyrus and L1 of the MEC similar to labelling after injection of AAV-FLEX-GFP into the LEC of *Sim1*[Cre] mice (*Figure 2*, *Figure 1B*). We observed

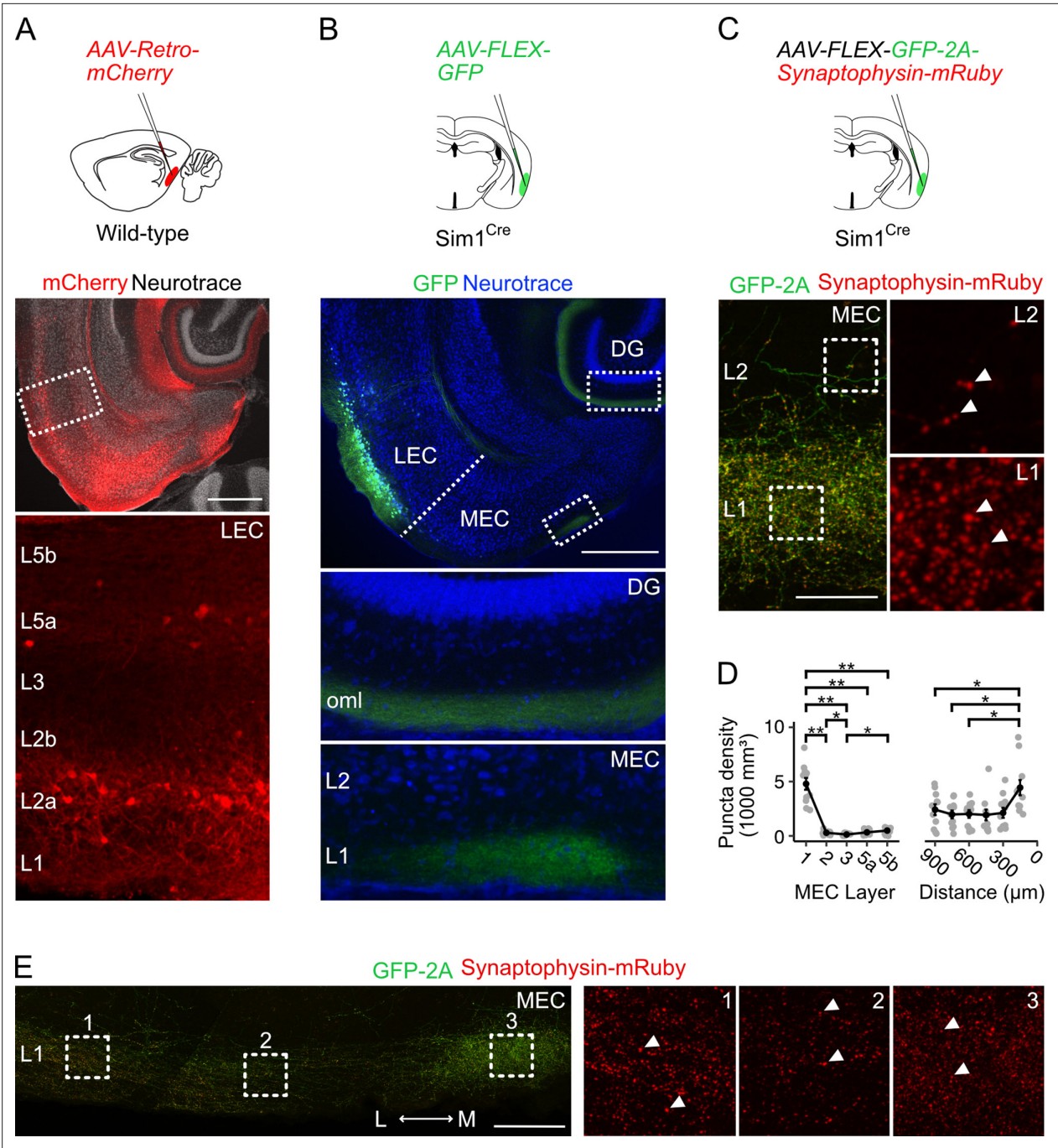

**Figure 1.** Fan cells in lateral entorhinal cortex send projections to medial entorhinal cortex that terminate in layer 1. (**A**) Schematic of strategy for targeting AAV-Retro-mCherry to the medial entorhinal cortex (MEC) of wild-type mice (top) and a horizontal brain section (bottom) showing retrograde labelling of neurons with mCherry in layers (L) 2a and 5a of lateral entorhinal cortex (LEC). Scale bar represents 500 µm. (**B**) Schematic of strategy for targeting AAV-FLEX-GFP to fan cells in *Sim1*^Cre mice (top) and horizontal brain section (bottom) showing labelling of fan cell axons with GFP (green) in the outer molecular layer (oml) of the dentate gyrus (DG) and L1 of the MEC. Neurons are counterstained with Neurotrace (blue). Scale bar represents 500 µm. (**C**) Schematic of strategy for targeting AAV-FLEX-GFP-2A-Synaptophysin-mRuby to fan cells in *Sim1*^Cre mice (top) and horizontal brain section (bottom) showing fan cell axons (green) and their terminals (red) in L1 and L2 of the MEC. Insets show synaptic puncta, indicated by white arrows. Scale bar represents 50 µm. (**D**) Line plots showing the volumetric density of axon terminals across L1-5b (left) and the mediolateral axis of the MEC (right). The mediolateral (M-L) axis of MEC was binned in 150 µm steps from the parasubiculum border. Black line shows population averages, and grey circles are values for single brain slices. Error bars represent SEM. There was a significant effect of MEC layer ($X^2_{(4)}$=32.8, p=0.000001, Kendall $W$=0.745, Friedman test) and location ($X^2_{(5)}$=21.6, p=0.0006, Kendall $W$=0.394) on puncta density. There was a higher density of puncta in L1 (4790±545 mm³) compared to all other layers (L2: $W$=66, p=0.01; L3: $W$=66, p=0.01; L5a: $W$=66, p=0.01; L5b: $W$=66, p=0.01, pairwise Wilcoxon tests with Bonferroni correction) and a

*Figure 1 continued on next page*

*Figure 1 continued*

higher density of puncta in L2 and L5b compared to L3 (L2: *W*=63, p=0.049; L5b: *W*=1, p=0.02). Further, there was a higher puncta density in the region of MEC closest to parasubiculum as compared to the regions laterally distanced 900 µm (*W*=66, p=0.015), 750 µm (*W*=66, p=0.015), and 600 µm (*W*=66, p=0.015). Asterisks indicate significance (**p<0.01, ***p<0.001, ****p<0.00001). (**E**) Same as **C**, but showing fan cell axons and axon terminals across the mediolateral axis of MEC L1. Scale bar represents 100 µm.

comparable levels of axonal labelling in both structures, with peak fluorescence in L1 of MEC and in the outer molecular layer of dentate gyrus (*Figure 2D–E*). Sections with the densest axonal labelling in the hippocampus and MEC were always dorsal to the labelled neurons at the injection site in L2a of LEC, suggesting that fan cell axons extend dorsally in the brain to synapse on postsynaptic neurons in dentate gyrus and MEC (*Figure 2F*). In doing so, the axons appear to either pass through L1 of LEC to reach superficial MEC directly, or may branch in deep MEC and the parasubiculum en route to the dentate gyrus (*Figure 2G*).

Together, these data show that fan cells in L2a of LEC send axonal projections to MEC that terminate in L1 with the greatest density of synapses at medial locations near the border with the parasubiculum. The projections to MEC arise as collaterals of axons that project to the dentate gyrus, with the axons potentially branching in the deep MEC and parasubiculum and projecting to regions of the MEC dorsal to their origin in the LEC.

## Fan cell projections provide a route for sensory and higher order cortical signals to reach the MEC

The LEC is well-positioned to integrate sensory and higher order signals because it receives long-range inputs from many neocortical structures (*Beckstead, 1978*; *Burwell and Amaral, 1998a*; *Burwell and Amaral, 1998b*; *Doan et al., 2019*; *Insausti et al., 1987*; *Johnson et al., 2000*; *Jones and Witter, 2007*; *Kerr et al., 2007*; *Kondo and Witter, 2014*; *Mathiasen et al., 2015*; *Naber et al., 1997*; *Suzuki and Amaral, 1994*; *Vaudano et al., 1991*). Many of these cortical inputs have axons that pass through L2, including inputs from medial prefrontal cortex (mPFC) (*Apergis-Schoute et al., 2006*; *Kondo and Witter, 2014*; *Room and Groenewegen, 1986*) and piriform cortex (PIR) (*Beckstead, 1978*; *Burwell and Amaral, 1998b*; *Johnson et al., 2000*; *Kerr et al., 2007*). In contrast, the superficial layers of the MEC appear to receive fewer direct neocortical inputs (*Cappaert et al., 2015*). Our finding of dense axonal projections from LEC to the superficial MEC (*Figure 1*, *Figure 2*) suggests a route by which cortical signals could reach neurons in the MEC indirectly.

To test whether projections from LEC fan cells to the MEC could relay information from cortical structures, we injected anterogradely transported AAV encoding Cre (pENN-AAV-hSyn-WPRE-Cre) into the mPFC or PIR of wild-type mice (*Figure 3A–B*, mPFC: n=4, PIR: n=4) and AAV mediating Cre-dependent expression of GFP (AAV-Flex-GFP) into the superficial LEC. This approach revealed labelling of fan cells in LEC (*Figure 3C and F*), which is consistent with their receiving direct synaptic input from the mPFC and piriform cortex (*Apergis-Schoute et al., 2006*; *Beckstead, 1978*; *Burwell and Amaral, 1998b*; *Johnson et al., 2000*; *Kerr et al., 2007*; *Kondo and Witter, 2014*; *Room and Groenewegen, 1986*). We observed axons of the labelled fan cells in the dentate gyrus and in L1 of MEC (*Figure 3D–E and G–H*), demonstrating that fan cells with postsynaptic targets in MEC receive direct cortical inputs. Thus, inputs from fan cells in the LEC may provide a route for sensory and higher order cognitive signals to reach the MEC.

## Activation of inputs from fan cells causes mixed excitation and inhibition of principal neurons in MEC

Our anatomical experiments demonstrate that LEC projections primarily terminate in L1 of MEC (*Figure 1C–E*). The ubiquity of fan cell axon terminals across the mediolateral axis of L1 suggests that fan cell projections could have widespread postsynaptic targets in the principal neuron populations residing in the deeper layers of MEC with dendrites that arborise in L1. To test whether specific populations of principal neurons receive projections from L2 of LEC, we enabled optogenetic activation of fan cell axons by injecting AAV mediating Cre-dependent expression of channelrhodopsin (AAV-FLEX-EF1a-DIO-hChR2(H134R)-EYFP) into the LEC of *Sim1*^Cre mice (n=67 mice). We then made whole-cell patch clamp recordings from principal neurons in ex-vivo brain slices containing the MEC (n=197 neurons) (*Figure 4A*, *Figure 4—source data 1*). We found that activation of fan cell inputs

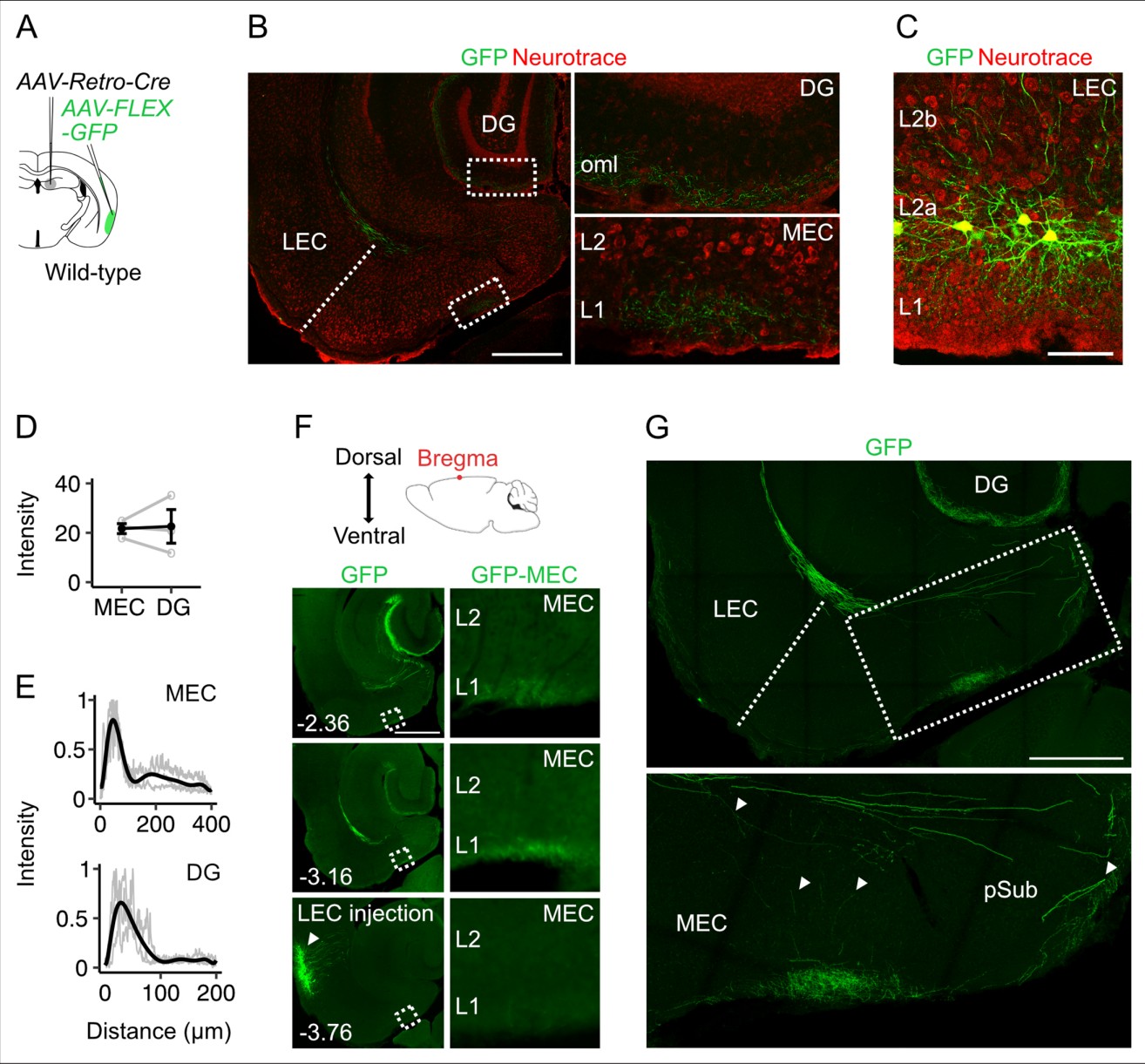

**Figure 2.** Projections to the medial entorhinal cortex from lateral entorhinal cortex arise from fan cells that also send projections to the hippocampus. (**A**) Schematic of strategy for targeting GFP specifically to fan cells in the lateral entorhinal cortex (LEC) that project to the hippocampal dentate gyrus (DG). In a wild-type mouse, adeno-associated virus (AAV) encoding retrograde Cre (AAV-Retro-Cre, grey) was injected into the DG and Cre-dependent AAV encoding GFP (AAV-FLEX-GFP, green) was injected into the LEC. (**B**) Horizontal brain section from a wild-type mouse showing GFP labelling of fan cell axons (green) in the outer molecular layer (oml) of the DG and layer (L) 1 of the medial entorhinal cortex (MEC). Neurons are counterstained with Neurotrace (red). Scale bar represents 500 µm. (**C**) Horizontal brain section from the same mouse shown in B, showing retrograde labelling of fan cell bodies with GFP (green) in L2a of the LEC. Scale bar represents 100 µm. (**D**) Fluorescence intensity of fan cell axons in L1 of MEC and the oml of DG. Intensity was quantified as the mean gray value of pixels in regions of interest (ROIs), where the possible range of values was 1–255 and a value of 255 corresponds to white. Mean intensity values were quantified for ROIs in MEC and DG, and were adjusted to baseline by subtracting the intensity value of an ROI from the same slice that did not contain fan cell axons. Fluorescence intensities were similar for DG and MEC ($W$=4, p=1, Mann Whitney U test). Black line shows the population average and grey circles are values for single brain slices. Error bars are SEM. (**E**) Fluorescence intensity as a function of distance across layers of MEC (top) and DG (bottom). For MEC, intensity was sampled from the edge of the slice (L1) towards the hippocampus. For DG, intensity was sampled from the outer edge of the oml towards the hilus. Plots are normalised to the minimum and maximum intensity values. Black line is a polynomial fit to the population data, and gray lines are values for single brain slices. (**F**) The dorsal-ventral position of the horizontal sections was estimated relative to bregma (top). Epifluorescent images of horizontal brain sections show GFP expression at dorsoventral locations containing the MEC and extending to the injection site in LEC. Zoomed in images of the ROIs highlighted by boxes in the left panels are shown in the panels to the right. Numbers indicate depth of slice from bregma in mm. Scale bar represents 500 µm. (**G**) Confocal images of a horizontal

*Figure 2 continued on next page*

Figure 2 continued

brain section showing fan cell axons expressing GFP in the MEC and parasubiculum (pSub)(upper). The zoomed in image (lower) shows the region highlighted by the rectangle, which contains axon collaterals in the deep MEC and the pSub. Putative branching points are indicated by white arrows. Scale bar represents 500 μm.

evoked postsynaptic membrane potential responses in principal neurons in all layers tested, revealing connectivity with a substantial proportion of stellate and pyramidal neurons in the superficial layers (L2: 127/158 cells, 80.4%, 64 mice; L3: 12/23 cells, 52.1%, 19 mice) and less numerous connections with pyramidal neurons in the deep layers (L5a: 2/11 cells, 18.2%, 10 mice; L5b: 1/5 cells, 20.0%, 4 mice) (*Figure 4B–D*).

To test whether, and how, responses to LEC inputs differed between cell types, we compared characteristics of the postsynaptic potentials. In L2 of the MEC, optogenetic stimulation of fan cell axons evoked both subthreshold and suprathreshold postsynaptic responses in stellate cells (L2 SC, 90/111 cells, 2 suprathreshold) and pyramidal cells (L2 PC, 37/47 cells, 2 suprathreshold). The typical subthreshold response to activation of fan cell inputs in L2 SCs and L2 PCs was biphasic, with an initial excitatory postsynaptic potential (EPSP) followed by an inhibitory postsynaptic potential (IPSP) (L2 SC: 58/90 cells, 64.4%; L2 PC: 21/37 cells, 56.8%), although some neurons demonstrated purely excitatory postsynaptic responses (L2 SC: 27/90 cells, 30.0%; L2 PC: 16/37 cells, 43.2%), and a smaller subset of L2 SCs demonstrated purely inhibitory postsynaptic responses (5/90 cells, 5.6%). In contrast, stimulation of fan cell axons evoked exclusively subthreshold postsynaptic responses in L3, L5a and L5b pyramidal neurons that were either biphasic (L3: 5/12 cells, 41.7%; L5a: 1/2 cells, 50.0%) or solely excitatory (L3: 7/12 cells, 58.3%; L5a: 1/2 cells, 50.0%; L5b: 1/1 cells, 100.0%). The amplitude of the EPSPs evoked after optogenetic stimulation of fan cell inputs to MEC differed between cell types (*Figure 4E*) being largest in L2 PCs on average compared to L2 SCs and principal cells in L3, L5a, and L5b. The amplitude of IPSPs, when present, did not differ significantly between cell types (*Figure 4E*). The latency of responses of L2 SCs and L2 PCs were generally short (L2 SC: 3.12±0.16ms; L2 PC: 3.97±0.68ms) suggesting they result from direct monosynaptic inputs (*Figure 4F*). In contrast, response latencies of L3 and L5 neurons were longer (*Figure 4F*), which could either result from filtering of responses propagating from distal dendritic regions, or could reflect a multisynaptic pathway.

Together, these data show that principal neurons in superficial and deep layers of the MEC respond to activation of inputs from LEC fan cells. Activation of these inputs can drive excitation and inhibition of circuits in the superficial and deep layers of MEC, with excitatory responses of pyramidal cells in layer 2 being substantially larger on average than responses of other principal cell types.

## Fan cells drive monosynaptic glutamatergic excitation and indirect GABAergic inhibition of layer 2 principal cells

Given the high probability of fan cell connectivity with L2 SCs and L2 PCs in comparison to principal neurons in the deeper layers of the MEC (*Figure 4C*), and the prevalence of spatial cells in this layer for which feature information would have functional relevance (*Gu et al., 2018*; *Høydal et al., 2019*; *Kitamura et al., 2015*; *Kitamura et al., 2014*; *Qin et al., 2018*; *Rowland et al., 2018*; *Tennant et al., 2018*), our further analyses focus on neurons with cell bodies in L2 of the MEC. First, we examined the receptor pharmacology of EPSPs evoked in L2 SCs and L2 PCs. The EPSPs evoked in both cell types were abolished by the AMPA receptor antagonist NBQX and NMDA receptor antagonist AP-5, indicating that they are glutamatergic (*Figure 5A–B*). Consistent with their reflecting direct monosynaptic activation, the EPSPs showed low variation in their latencies (*Figure 5C–D*), and following abolition by application of tetrodotoxin (TTX) could be recovered by further application of 4-aminopyridine (4-AP; *Figure 5E–F*).

Inhibitory components of responses to optogenetic activation of fan cell axons usually occurred immediately following an EPSP, and were also abolished by glutamatergic antagonists, indicating they require excitation of intermediary neuronal populations (*Figure 5A–B*). The GABA$_A$ receptor antagonist Gabazine abolished the fast rising inhibitory response but left intact a slower inhibitory potential (*Figure 6A*). This slower potential was subsequently abolished by the GABA$_B$ receptor antagonist CGP55845 (*Figure 6A*). Conversely, in separate experiments in which GABA$_B$ receptors were blocked first, the slow IPSP was abolished, while the fast IPSP was left intact and subsequently blocked by GABA$_A$ receptor antagonists (*Figure 6B*). Bath application of GABA receptor antagonists

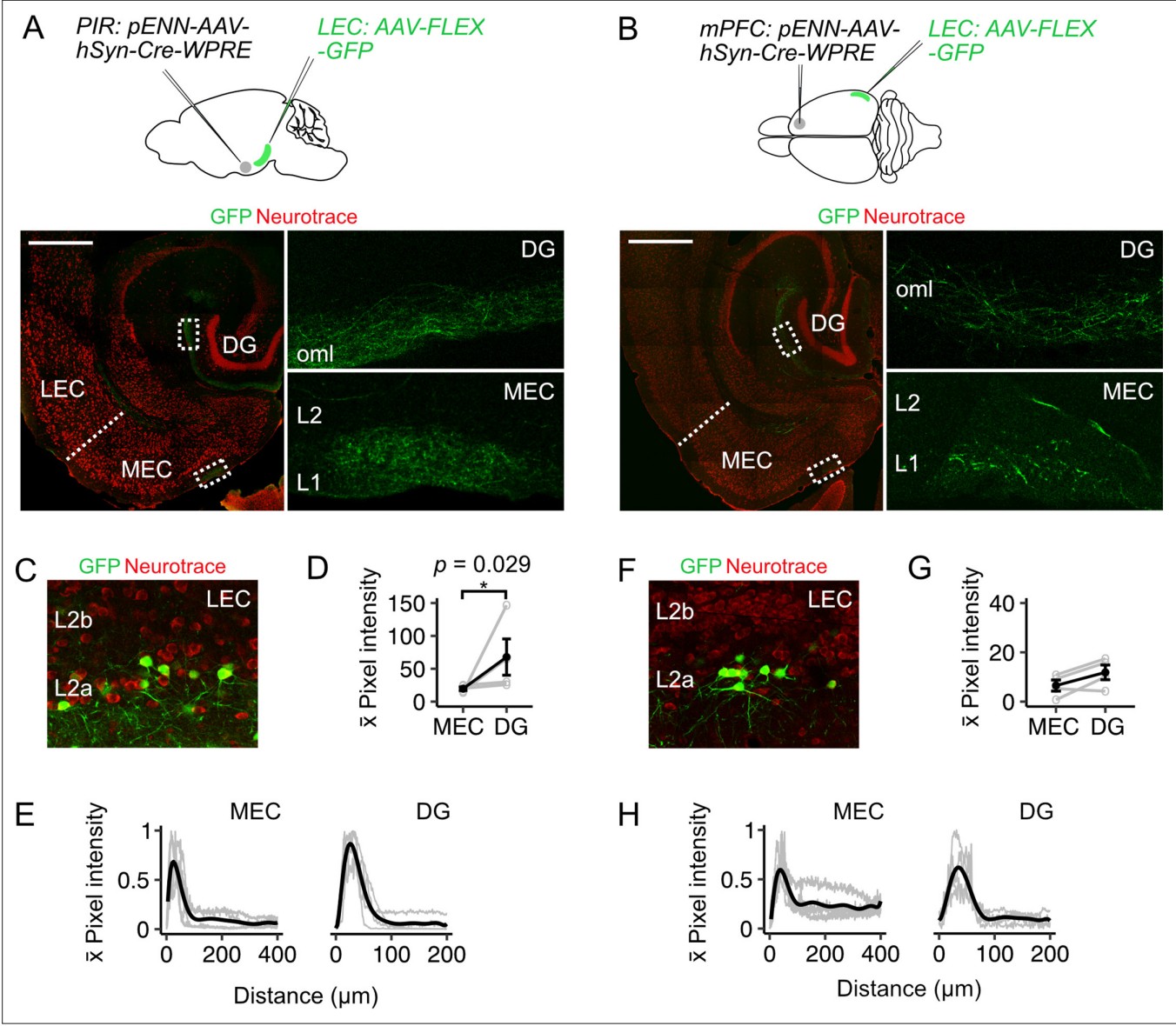

**Figure 3.** Fan cells that project to medial entorhinal cortex receive inputs from piriform and prefrontal cortices (**A-B**) (Top). Schematic of strategy for targeting GFP specifically to fan cells in the lateral entorhinal cortex (LEC) that receive projections from piriform cortex (PIR, A) or medial prefrontal cortex (mPFC, **B**). AAV that anterogradely transports Cre (pENN-AAV-hSyn-Cre-WPRE, grey) was injected into the PIR or mPFC of a wild-type mouse. Cre-dependent AAV encoding GFP (AAV-FLEX-GFP, green) was injected into the LEC of the same mouse to label Cre-expressing fan cells and their axons. Horizontal brain section showing labelling of fan cell axons with GFP in the outer molecular layer (oml) of the dentate gyrus (DG) and layer (L) 1 of the MEC (right panels show zoomed in images of the regions of interest highlighted in the left panels). Scale bar is 500 µm. (**C**) Expression of GFP in fan cells at the injection site in the same animal as A. Neurons are counterstained with Neurotrace (red). (**D**) Fluorescence intensity in L1 of MEC and the oml of DG of axons from fan cells receiving inputs from PIR. The fluorescence intensity of fan cell axons was higher in the DG (*W*=16, p=0.029, Mann Whitney U test). Black line shows population average and grey circles are values for single brain slices. Error bars are SEM. (**E**) Fluorescence intensity of axons from fan cells that receive PIR inputs as a function of distance across layers of MEC (left) and DG (right). For MEC, intensity was sampled from the edge of the slice (L1) towards the hippocampus. For DG, intensity was sampled from the outer edge of the oml towards the hilus. ROIs were oriented perpendicular to the layers of MEC and DG. Plots are normalised to the minimum and maximum intensity values. Black line is population data fit with a polynomial model and gray lines are values for single brain slices. (**F**) Same as C but for fan cells labelled on the basis of receiving projections from medial prefrontal cortex (mPFC). (**G-H**) Same as (**D-E**), but for axons from fan cells that receive inputs for mPFC. Fluorescence intensities of axons from fan cells that receive inputs from mPFC was similar for DG and MEC (*W*=12, p=0.343).

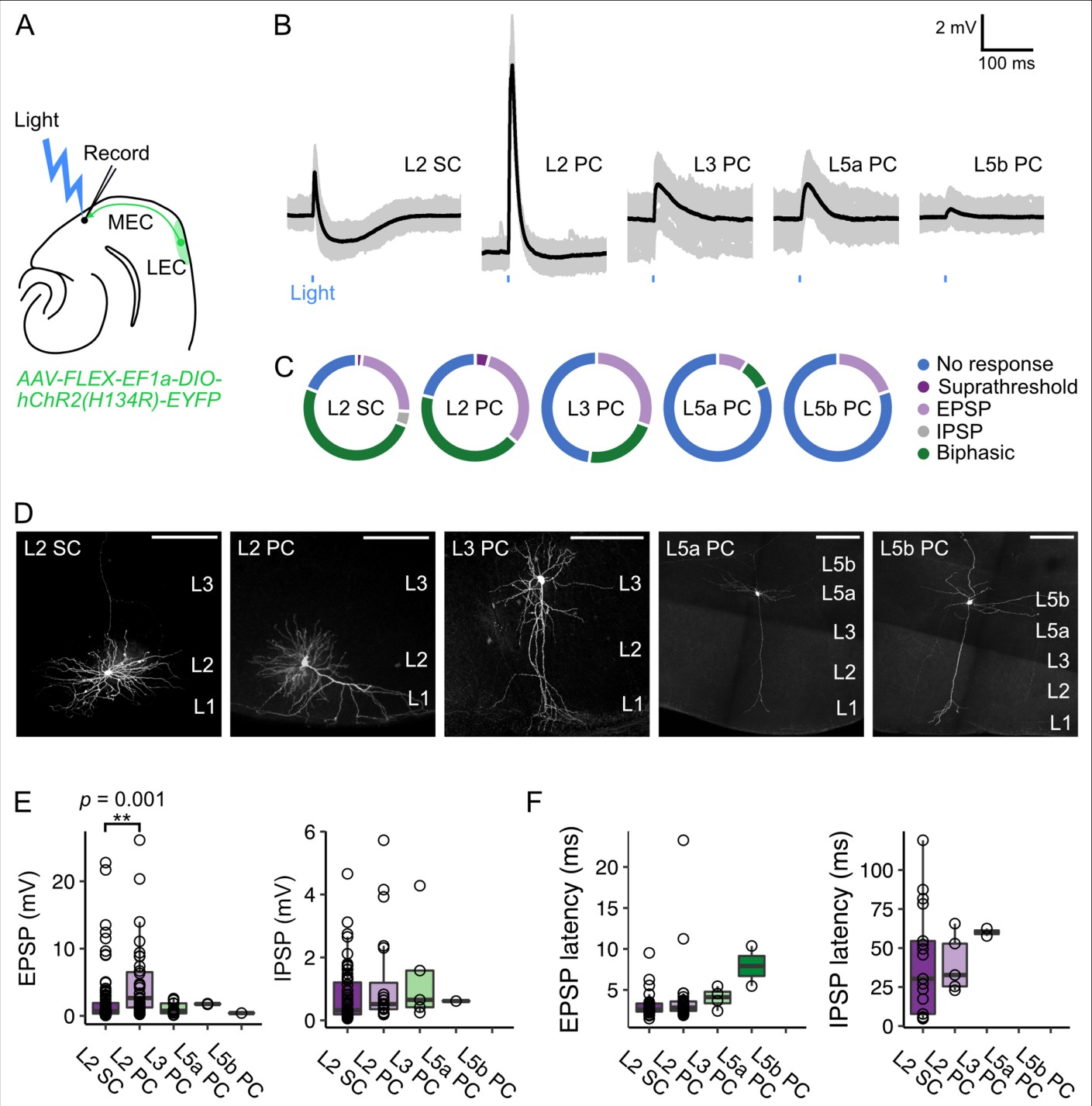

**Figure 4.** Optogenetic activation of fan cell inputs evokes postsynaptic responses in medial entorhinal cortex principal neurons. (**A**) Schematic of recording experiment to evaluate the postsynaptic responses in medial entorhinal cortex (MEC) to stimulation of fan cell axons. AAV-EF1a-DIO-ChR2(H134R)-EYFP was injected into the LEC of *Sim1*Cre mice to enable optical activation of fan cells. Synaptic output from fan cells was evaluated by recording light-evoked responses of principal neurons in different layers of MEC. (**B**) Representative examples of membrane potential responses evoked by optical activation of fan cell axons in principal neurons in layers (**L**) 2, 3, 5a, and 5b of the MEC (SC = stellate cell, PC = pyramidal cell). Blue bar indicates the 3 ms period of stimulation. Individual traces (grey) are overlaid with an average of all traces (black). (**C**) Charts showing the proportion of each type of neuron that demonstrated different types of membrane potential responses to activation of fan cell inputs, including no response (blue), suprathreshold (dark purple), subthreshold excitatory (EPSP, light purple), inhibitory (IPSP, grey), and biphasic (green). (**D**) Representative examples of neurons that were recorded from each layer of MEC and filled with Biocytin (white) in horizontal brain sections. Scale bars represent 200 µm. (**E**) Boxplots showing amplitudes for excitatory (EPSP) and inhibitory (IPSP) membrane potential responses to activation of fan cell inputs in principal

*Figure 4 continued on next page*

*Figure 4 continued*

neurons in L2, L3, L5a, and L5b of the MEC. Circles represent values for individual neurons. Comparison of response amplitudes across cell-types using a Kruskall-Wallis test revealed a significant effect of cell type on EPSP amplitude ($H_{(4)}$ = 17.261, p=0.002, $\eta^2$=0.097), but not IPSP amplitude ($H_{(4)}$ = 3.624, p=0.305). Post-hoc pairwise comparisons revealed that EPSPs were larger in L2 PCs compared to L2 SCs (Z=3.89, p=0.001; *Figure 4—source data 2*). (**F**) Boxplots showing response latencies of principal neurons in L2, L3, L5a, and L5b with an average membrane response amplitude of > 1 mV. Latency was measured as the time from stimulus onset to 10% deviation from baseline membrane potential. There was no effect of cell type on the latency of EPSPs or IPSPs from stimulus (EPSP: $H_{(3)}$ = 7.550, p=0.056, IPSP: $H_{(2)}$ = 1.663, p=0.435).

The online version of this article includes the following source data for figure 4:

**Source data 1.** Summary of subthreshold and suprathreshold membrane properties of recorded neurons.

**Source data 2.** Summary of pairwise comparisons of membrane potential responses.

had a modest effect on EPSP amplitudes (*Figure 6A–B*) but substantially increased the half-width of the excitatory potentials, suggesting that fast inhibition is primarily acting to control the duration of excitation rather than modulate its amplitude.

Together, these experiments demonstrate that EPSP responses of L2 principal neurons to input from fan cells are mediated by monosynaptic glutamatergic excitation. In contrast, IPSP responses are indirect and are consistent with glutamatergic excitation of interneuron populations; this could in principle arise from feedforward excitation of interneuron populations by fan cell inputs or by recurrent

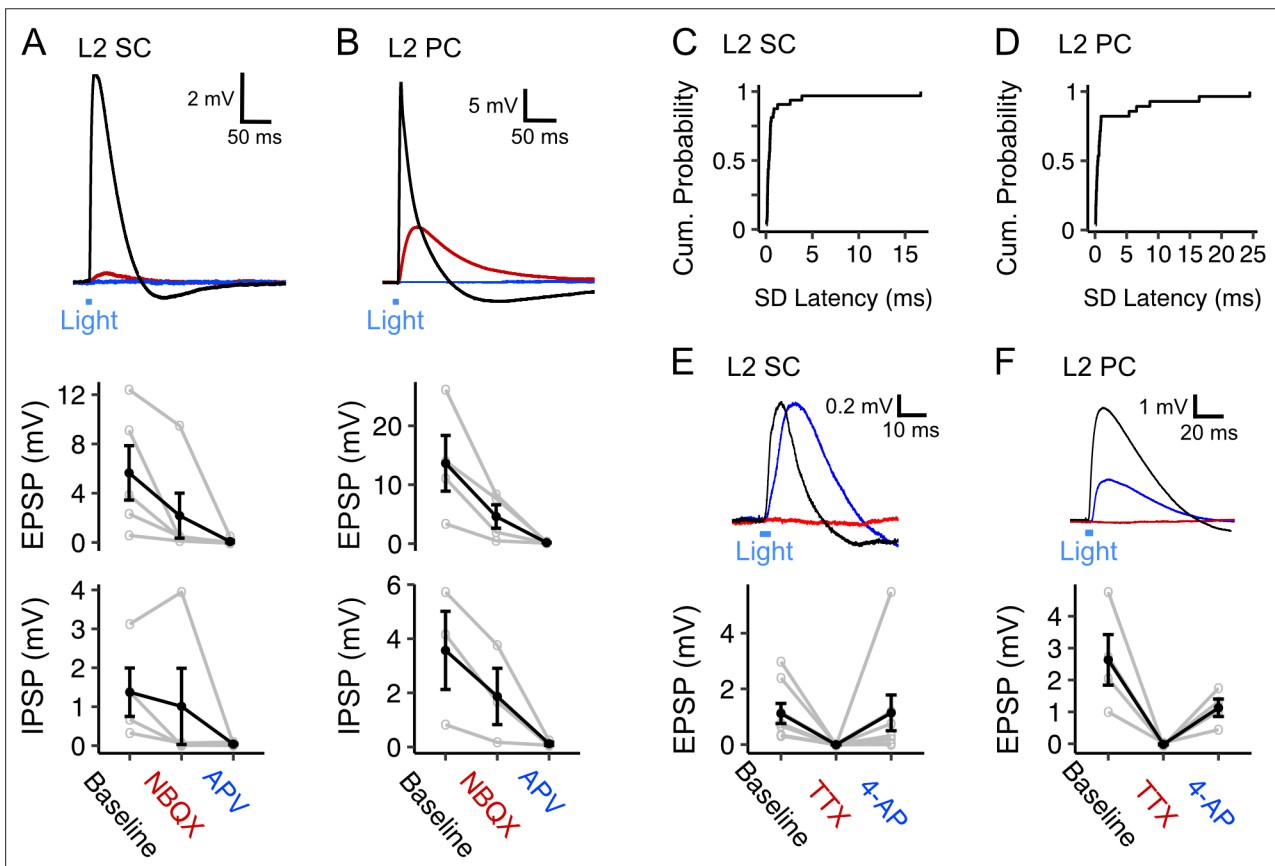

**Figure 5.** Fan cell inputs to stellate and pyramidal cells in layer 2 of the medial entorhinal cortex are glutamatergic and monosynaptic (**A, B**) (Top) Membrane potential response of a layer 2 stellate cell (L2 SC, **A**) and a layer 2 pyramidal cell (L2 PC, **B**) after optogenetic activation of fan cell inputs. The membrane potential response was abolished by application of ionotropic glutamate receptor antagonists NBQX (10 µM, red) and APV (50 µM, blue). Blue line indicates the 3 ms period of optical stimulation. (Bottom) Quantification of the light-evoked membrane potential response in stellate (n=5) and pyramidal cells (n=4) after application of NBQX and APV. Data is shown for excitatory (EPSP) and inhibitory (IPSP) components. Black lines indicate population average and grey lines indicate individual neurons. Error bars are SEM. (**C-D**) Cumulative probability of the standard deviation of EPSP latencies for layer 2 stellate (C, $\bar{x}$=1.04 ± 0.312, IQR = 0.311) and pyramidal cells (D, $\bar{x}$=2.55 ± 0.918, IQR = 0.693). (**E-F**) (Top) Similar to (**A-B**), but showing effects on a layer 2 stellate (E) and pyramidal cell (**F**) of application of TTX (500 nM, red) and its recovery with application of 4-AP (200 µM, blue). (Bottom) Quantification of light-evoked EPSPs in stellate (n=8) and pyramidal cells (n=4) after application of TTX and 4-AP.

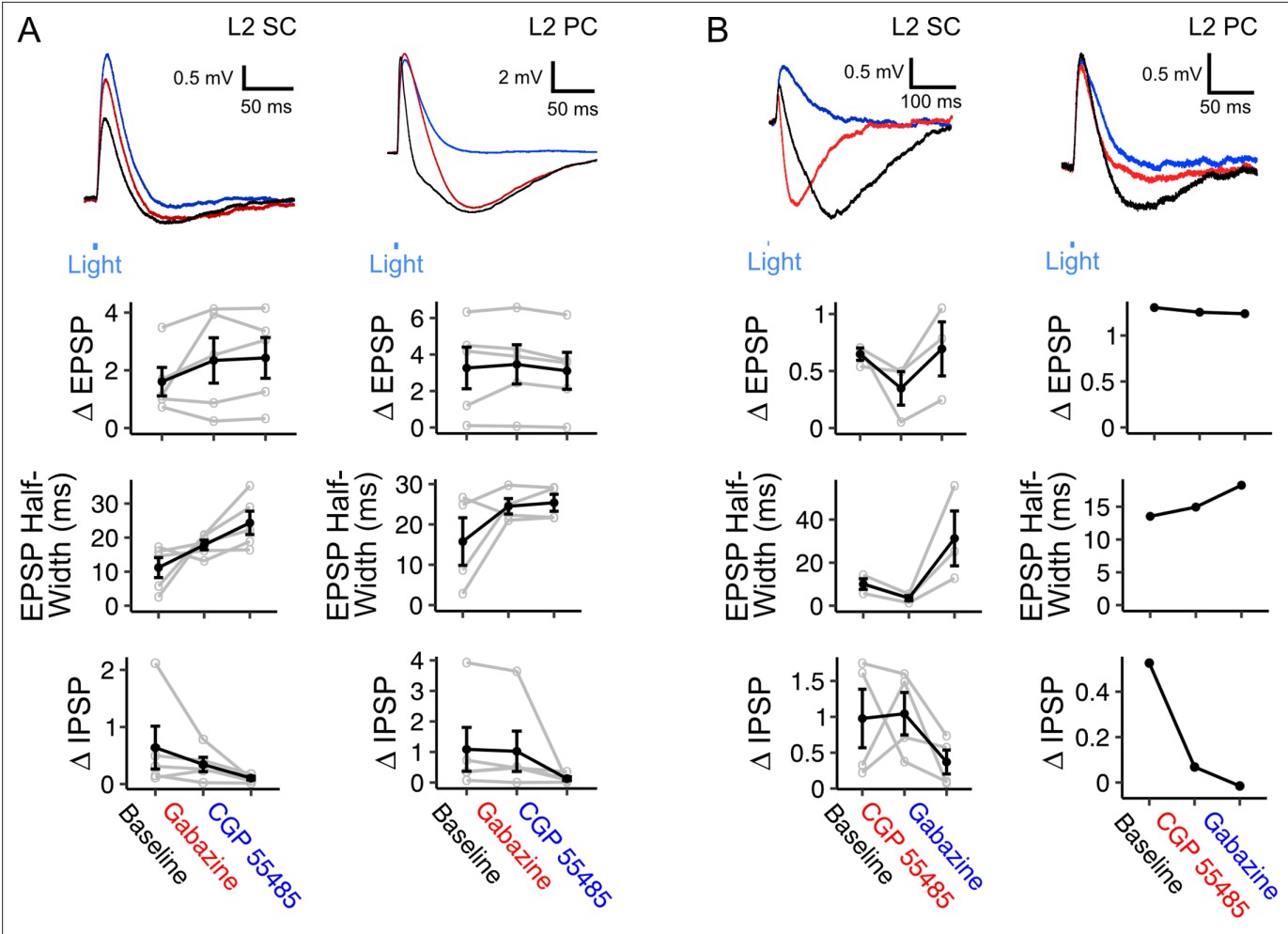

**Figure 6.** Inhibition of principal neurons in medial entorhinal cortex by inputs from fan cells is mediated by GABA_A and GABA_B receptors. (**A**) (Top) Membrane potential response of a layer 2 stellate cell (L2 SC, left) and pyramidal cell (L2 PC, right) after optogenetic activation of inputs from fan cells. Different inhibitory components of the membrane potential response were abolished by application of GABA_A receptor antagonist Gabazine (10 µM, red) and GABA_B receptor antagonist CGP55845 (100 µM, blue). Blue line indicates the 3 ms period of optical stimulation. (Bottom) Quantification of the light-evoked membrane potential response in layer 2 stellate (n=5) and pyramidal cells (n=5) after application of Gabazine and CGP55845. Amplitude measurements are shown for excitatory (EPSP) and inhibitory (IPSP) postsynaptic potentials and half-width measurements are also shown for EPSPs. At a population level, GABA receptor antagonists causes a significant change in the halfwidth of EPSPs for L2 SCs ($X^2_{(2)}$=8.4, p=0.015, Friedman test), but not L2 PCs ($X^2_{(2)}$=1.2, p=0.472), and did not have significant effects on E/IPSP amplitudes for either cell type (L2 SC: EPSP, $X^2_{(2)}$=2.8, p=0.247; IPSP: $X^2_{(2)}$=5.2, p=0.074; L2 PC: EPSP, $X^2_{(2)}$=4.8, p=0.091; IPSP: $X^2_{(2)}$=5.2, p=0.074). Analyses at the level of individual cells revealed effects of antagonist application on measurements of E/IPSP amplitude and EPSP halfwidth for almost all neurons (*Figure 6—source data 1*, *Figure 6—source data 2*, *Figure 6—source data 3*). Black lines indicate population average and grey lines indicate an individual neuron. Error bars are SEM. (**B**) Same as **A**, but showing application of GABA_B receptor antagonist CGP55845 (red) followed by application of GABA_A receptor antagonist Gabazine (blue) for layer 2 stellate (n=4) and pyramidal cells (n=1). At a population level GABA-receptor antagonists caused significant changes in the halfwidth of EPSPs ($X^2_{(2)}$=6.0, p=0.0498), but did not their amplitude($X^2_{(2)}$=4.67, p=0.097) or IPSPs ($X^2_{(2)}$=2.67, p=0.264). As for A, analysis for individual neurons revealed a significant effect of GABA-receptor antagonists on measurements of E/IPSP amplitude and EPSP half-width for almost all neurons (*Figure 6—source data 3*).

The online version of this article includes the following source data for figure 6:

**Source data 1.** Summary of cell-level analysis of effects of GABA receptor antagonists on synaptic response properties.

**Source data 2.** Summary of cell-level pair-wise comparisons of effects of GABA receptor antagonists on synaptic response properties.

**Source data 3.** Summary of cell-level pair-wise comparisons of effects of GABA receptor antagonists on synaptic response properties.

feedback excitation following spiking by L2 principal cells. The later onset of inhibitory responses and the relatively small effect of GABAergic block on EPSP amplitudes suggests that fast GABA$_A$ receptor mediated inhibition in this pathway may function to determine the duration of excitatory responses, whereas the slower GABA$_B$ receptor mediated inhibition may serve to make neurons refractory to subsequent inputs.

## Interneurons in L1 of MEC mediate slow components of inhibitory input to layer 2 principal neurons

To test whether the connectivity of inputs from fan cells can support feedforward inhibition within the MEC we examined responses of interneuron populations in the superficial MEC to optogenetic activation of fan cell axons (n=59 cells, 34 mice). We evaluated responses of interneurons in L2 (42 cells, 30 fast-spiking), which have well established connectivity with L2 principal cell populations (*Couey et al., 2013*; *Nilssen et al., 2018*; *Pastoll et al., 2013*), and interneurons in L1, which to date have received very little attention.

Optogenetic activation of fan cell axons evoked postsynaptic membrane potential responses in all interneurons recorded in L1 (17/17 cells, 100.0%) and the majority of interneurons in L2 (32/42 cells, 76.2%) (*Figure 7A–D*). In L1 and L2 interneurons, stimulation of fan cells axons evoked a mix of suprathreshold and subthreshold responses (L1: 4/14 cells suprathreshold, 28.6%; L2: 7/32 cells suprathreshold, 21.9%). In L2, stimulation of fan cell axons evoked responses in low-threshold spiking interneurons (7/12, 58.3%, 2 suprathreshold) and fast-spiking interneurons (25/30, 83.3%, 5 suprathreshold). The relatively high prevalence of suprathreshold responses is consistent with L1 and L2 interneuron populations contributing to feedforward inhibition of principal cells. Postsynaptic responses in L1 were exclusively excitatory, and although most responses in L2 were also excitatory, one interneuron in L2 had a biphasic response containing excitatory and inhibitory components. EPSPs evoked in L1 and L2 interneurons had similar amplitudes and short latencies from stimulus onset (*Figure 7B*). The EPSPs were abolished after application of NBQX and AP-5 (*Figure 7E*), indicating that they are glutamatergic, and abolished in the presence of TTX but recovered after application of 4-AP, indicating that they result from direct monosynaptic activation (*Figure 7F*).

These data suggest that inhibition of L2 SCs and L2 PCs by fan cell inputs could be through feedforward pathways involving interneurons in L1 and L2 of the MEC. In support of this circuit model, it is well established that activation of interneurons in L2 generates fast IPSPs in nearby principal cells (*Couey et al., 2013*; *Fuchs et al., 2016*; *Pastoll et al., 2013*). However, responses of layer 2 principal cells to activation of L2 interneurons appear brief and so are unlikely to account for slow inhibitory components of responses to fan cell activation. To test whether slow inhibitory components could be mediated by interneurons in L1 of the MEC, we made simultaneous whole-cell patch clamp recordings from pairs of L2 SCs (n=22) or L2 PCs (n=5) and L1 interneurons (*Figure 8A–C*). We found direct inhibitory connections between L1 interneurons and both cell types (*Figure 8B*; L2 SC: 6/22 pairs, 27.3%; L2 PC: 2/5 pairs, 40.0%), where firing of an action potential from the L1 interneuron evoked an inhibitory postsynaptic potential in adjacent L2 SCs and L2 PCs (*Figure 8D–F*). These connections were unidirectional, with action potentials in L2 SCs or L2 PCs having no effect on the membrane potential of simultaneously recorded L1 interneurons. The time course of the IPSPs evoked in L2 SCs and L2 PCs following activation of L1 interneurons typically had two components, with a second relatively slow component having a time course comparable to the slow component of IPSPs evoked by activation of fan cell inputs (*Figure 8E–F*).

Together, these data demonstrate that interneurons in L1 and L2 of the MEC receive powerful excitatory input from LEC fan cells that is effective at driving spiking. We also show that L1 interneurons mediate slow inhibitory inputs to layer 2 principal cells in the MEC. Thus, our data support a circuit model in which fan cells drive feedforward inhibitory input to principal cells in layer 2, with L1 interneurons mediating slow inhibitory components.

## Fan cells inputs to pyramidal cells are dominated by excitation whereas responses of stellate cells are heterogeneous and independent of anatomical location

As we found considerable variation between neurons in the amplitudes of EPSPs and IPSPs evoked by fan cell activation, we wondered if there was any systematic relationship between the relative

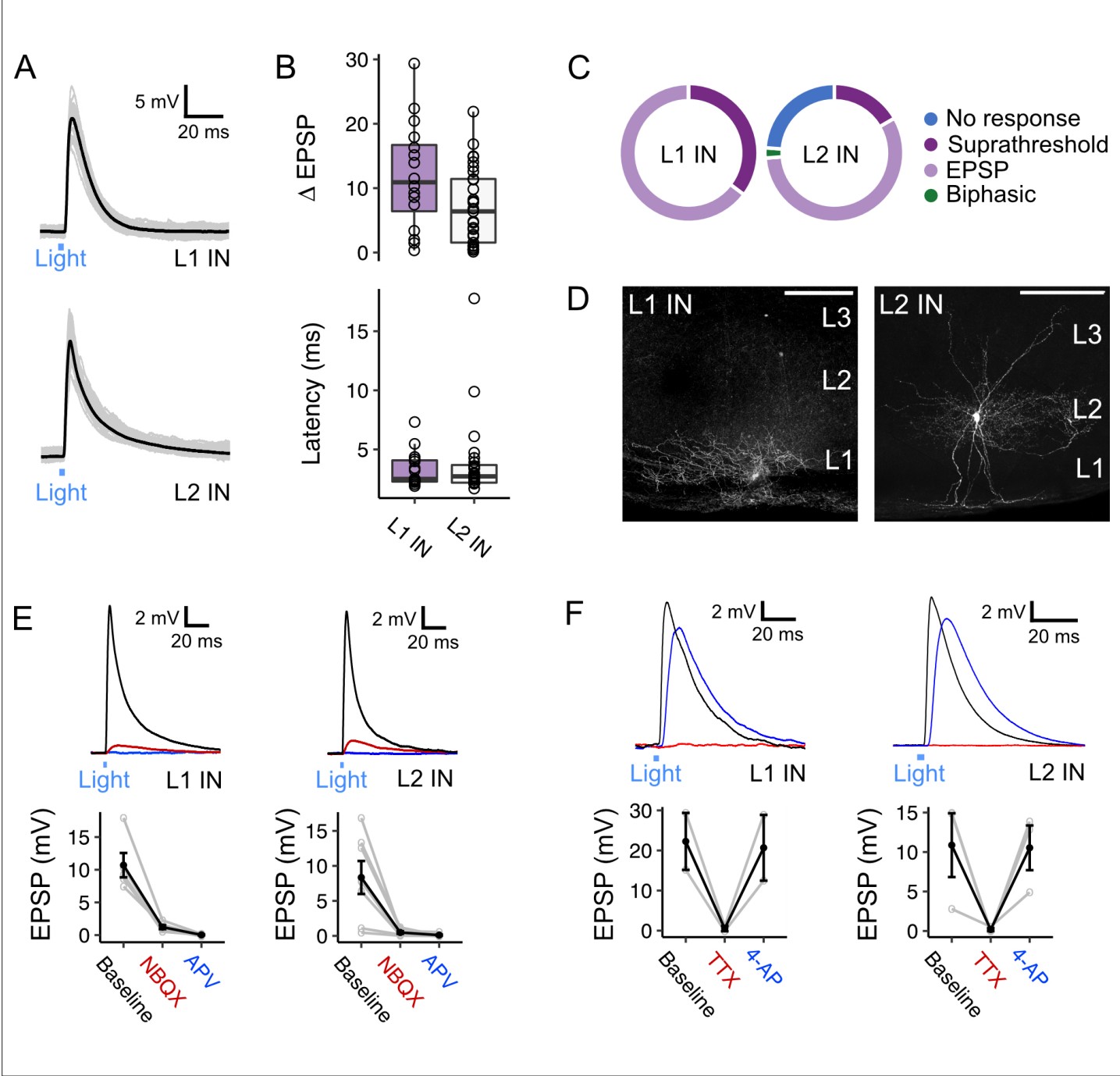

**Figure 7.** Activation of fan cell inputs to medial entorhinal cortex evokes monosynaptic glutamatergic postsynaptic responses in layer 1 and layer 2 interneurons. (**A**) Representative examples of membrane potential responses evoked by optogenetic activation of fan cell inputs for interneurons in layer 1 (L1 IN, top) and layer 2 (L2 IN, bottom) of medial entorhinal cortex (MEC). Blue bar indicates the 3 ms period of optical stimulation. Individual traces (grey) are overlaid by an average of all traces (black). (**B**) Boxplots showing the amplitude (top) and latency (bottom) of excitatory responses (EPSP) for interneurons in L1 (purple) and L2 (white). Circles represent single neurons. There was no difference in the amplitude (*W*=314, p=0.053, Mann Whitney U test) or latency (*W*=185, p=0.898) across cell-types. (**C**) Charts showing the proportion interneurons in L1 (left) and L2 (right) that demonstrated different types of responses to activation of fan cell inputs, including no response (blue), suprathreshold (dark purple), subthreshold excitatory (light purple) and biphasic (green). Low-threshold spiking (LTS) and fast-spiking (FS) interneurons both responded to stimulation of fan cell inputs (LTS: 7/12 cells, 2 suprathreshold; FS: 25/30 cells, 5 suprathreshold). (**D**) Representative examples of interneurons in L1 (left) and L2 (right) that were filled with biocytin (white) in horizontal brain sections. Scale bars represent 200 μm. (**E**) (Top) Membrane potential response of interneurons in L1 (left) and L2 (right) after optogenetic activation of fan cell inputs. The membrane response was abolished by application of ionotropic glutamate receptor antagonists NBQX

*Figure 7 continued on next page*

*Figure 7 continued*

(10 µM, red) and APV (50 µM, blue). Blue line indicates the 3 ms period of optical stimulation. (Bottom) Quantification of light-evoked EPSPs in L1 (n=7) and L2 interneurons (n=6) after application of NBQX and APV. Black lines indicate population average and grey lines indicate individual neurons. Error bars are SEM. (**F**) Same as **E**, but showing abolishment of EPSPs evoked in an L1 (left, n=2) and L2 (right, n=3) interneuron by application of TTX (500 nM, red) and their recovery with 4-AP (200 µM, blue).

strengths of each input component. E-I ratios, calculated as the amplitude of the EPSP normalised to the total amplitude of the EPSP and IPSP, were greater on average for L2 PCs compared to L2 SCs (*W*=1233.5, p=0.034, Wilcoxon test), with almost all L2 PCs showing a greater net excitation, whereas the L2 SC population was more heterogeneous and included neurons that showed greater net inhibition as well as neurons that favoured excitation (*Figure 9A*). For L2 SCs this variation was independent of the baseline membrane potential, indicating that it could not be explained by differences between neurons in the driving force for inhibitory responses (*Figure 9—figure supplement 1A*). For both L2 SCs and L2 PCs the greatest variation in E-I ratios was for EPSPs with amplitudes <10 mV (*Figure 9B–C*). This may reflect saturating activation of interneurons when more fan cell axons are activated, or saturated hyperpolarization as the membrane potential approaches the reversal potential for GABA-mediated responses. Thus, variation in excitatory and inhibitory input strength is a feature of input pathways from LEC to L2 SCs in the MEC. This may be of potential functional importance, for example to differentially recruit and suppress activation of neurons in the MEC.

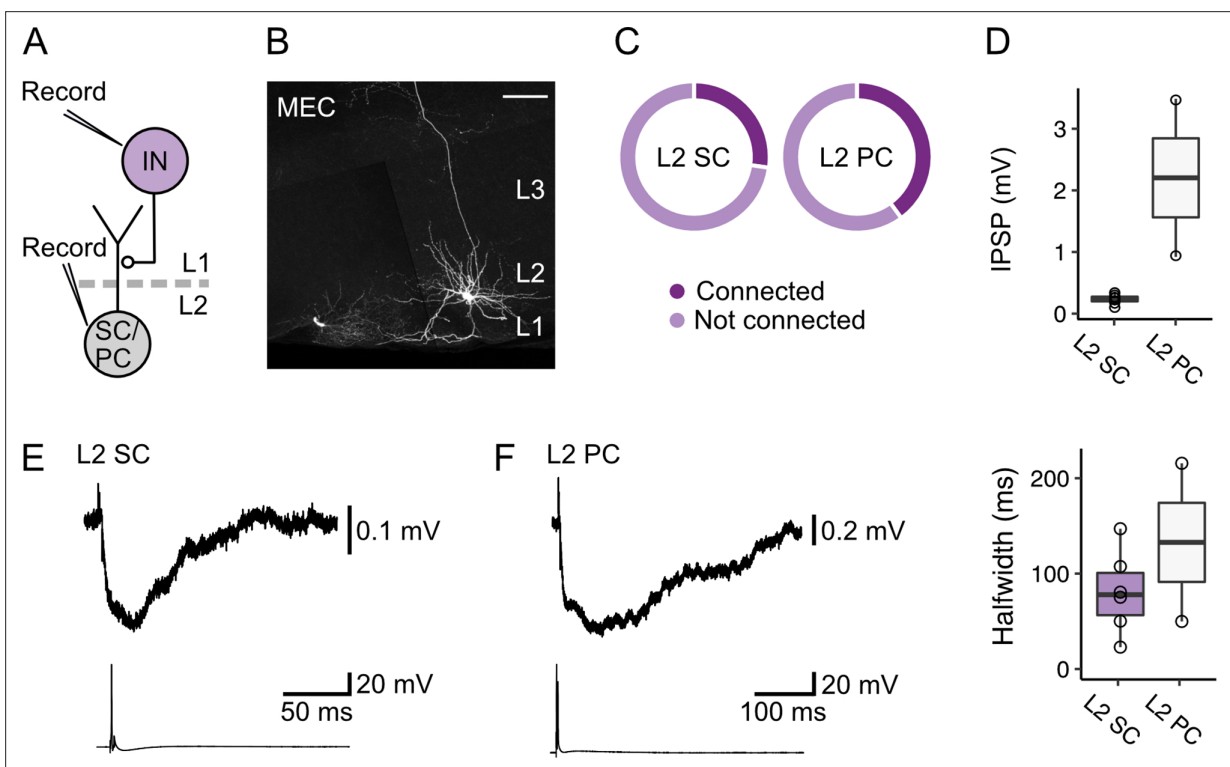

**Figure 8.** Layer 1 interneurons in the medial entorhinal cortex directly inhibit stellate and pyramidal cells in layer 2. (**A**) Schematic of recording configuration to evaluate postsynaptic responses in medial entorhinal cortex (MEC) layer 2 (L2) stellate (SC) and pyramidal cells (PC) to stimulation of interneurons (IN) in layer 1 (L1). Connectivity between L1 interneurons and principal cells was established with simultaneous measurement of membrane potentials in L2 stellate or pyramidal cells and activation of an interneuron in L1. (**B**) Representative image of a slice containing a simultaneously recorded interneuron in L1 (lower left) and an L2 stellate cell (right) in the MEC. Both recorded neurons were filled with biocytin (white). Scale bar represents 100 µm. (**C**) Charts show proportions of stellate (left) and pyramidal cells (right) that were hyperpolarized by (connected, dark purple) or did not demonstrate a membrane potential response (not connected, light purple) to activation of inputs from an interneuron in L1 of the MEC. (**D**) Quantification of the amplitude (top) and half-width (bottom) of inhibitory membrane potential response (IPSP) in layer 2 stellate (purple) and pyramidal cells (white) to activation of inputs from an interneuron in L1. Circles represent single neurons. (**E-F**) Representative traces of membrane potentials during the stimulation experiments for stellate (**E**) and pyramidal cells (**F**). Traces from the stellate or pyramidal cell are at the top, and action potentials fired by the interneuron in L1 with injection of current are shown on the bottom.

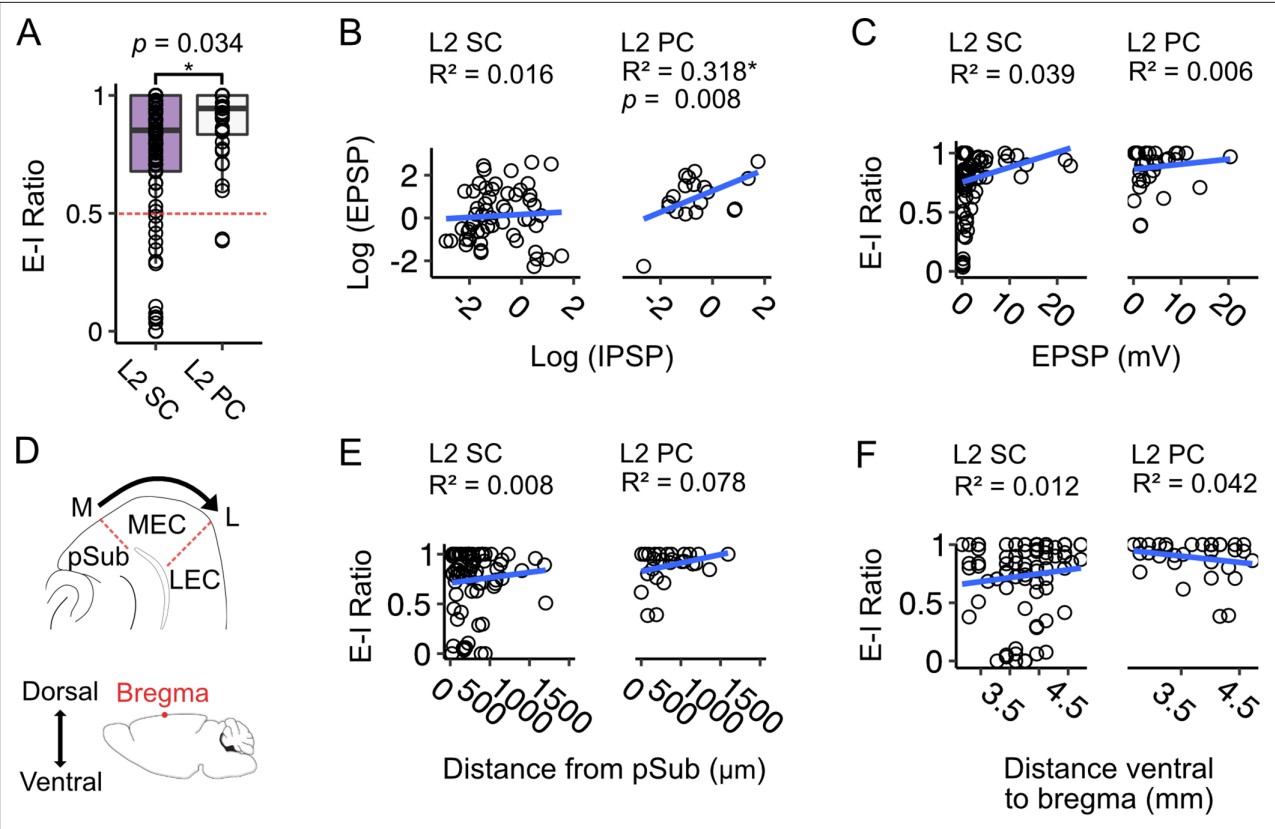

**Figure 9.** Excitation-inhibition bias of responses to fan cell inputs differs between stellate and pyramidal cells, is highly variable and is independent of location. (**A**) Boxplot comparing ratios of excitation to inhibition (E-I ratio) = EPSP amplitude/(EPSP amplitude +IPSP amplitude) for responses of layer 2 stellate (L2 SC, purple) and pyramidal cells (L2 PC, white) to activation of fan cell inputs. Dashed red line indicates equal amplitudes of excitation and inhibition. The bias to excitation was larger in pyramidal cells ($W$=1233.5, p=0.034, Mann Whitney U test). Each dot represents a single neuron. (**B**) EPSP amplitude as a function of IPSP amplitude for stellate (right) and pyramidal cells (left). Raw values were log transformed (see *Figure 9—figure supplement 1B* for raw data). EPSP amplitude increased with IPSP amplitude for pyramidal cells ($F_{(1,19)}$ = 8.877, p=0.008, F-test of overall significance), but not stellate cells ($F_{(1,57)}$ = 0.894, p=0.348). Blue line is fit of a linear regression model with the equation $y = \beta 0 + \beta 1x + \epsilon$. (**C**) Same as **B**, but with E-I ratio plotted as a function of EPSP amplitude. We did not find a relationship between E-I ratio and EPSP amplitude for either cell type (L2 SC: $F_{(1,84)}$ = 3.365, p=0.070; L2 PC: $F_{(1,34)}$ = 0.193, p=0.665). (**D**) Schematics illustrating estimation of the mediolateral (M-L, top) and dorsoventral (bottom) positions of neurons within the medial entorhinal cortex (MEC). On the mediolateral axis, neuron position was measured as distance of the cell body from the MEC border with parasubiculum (pSub), where larger values indicate proximity to the lateral border of MEC. On the dorsoventral axis, neuron position was measured as distance of the cell body from bregma (indicated by a red dot). (**E-F**) E-I ratio plotted as a function of mediolateral (**E**) or dorsoventral (**F**) position for stellate (left) and pyramidal cells (right). We did not find a relationship between E-I ratio and position for either cell type (Mediolateral position, L2 SC: $F_{(1,72)}$ = 0.564, p=0.455, L2 PC: $F_{(1,26)}$=2.203, p=0.150; Dorsoventral position, L2 SC: $F_{(1,77)}$ = 0.901, p=0.346, L2 PC: $F_{(1,31)}$=1.352, p=0.254).

The online version of this article includes the following figure supplement(s) for figure 9:

**Figure supplement 1.** Relationships between excitatory and inhibitory components and the resting membrane potential or position of neuron in medial entorhinal cortex.

**Figure supplement 2.** Relationships between response amplitudes in layer 1 and layer 2 interneurons and neuron position in medial entorhinal cortex.

Given that an L2 SC's dorsoventral position determines its intrinsic electrophysiological properties (*Boehlen et al., 2010*; *Garden et al., 2008*; *Giocomo et al., 2007*; *Pastoll et al., 2020*) and responses to inhibitory input (*Beed et al., 2013*), we asked if variation in E-I ratios is also related to neuronal location within the MEC. To explore this possibility, we assigned each neuron to a coordinate ventral to bregma by comparing the encompassing slice to an atlas of the mouse brain, and measured the distance of soma from the border of MEC with the parasubiculum (*Figure 9D*). We then applied linear models to quantify relationships between neuron location, E-I ratios and amplitude of excitatory and inhibitory postsynaptic potentials (*Figure 9E–F*; *Figure 9—figure supplement 1*).

For L2 SCs and L2 PCs, the E-I ratio of the membrane potential responses to activation of fan cell inputs did not vary with neuronal location on the mediolateral or dorsoventral axis of MEC

(*Figure 9E–F*), suggesting that the bias towards excitation we find in both cell-types is not a function of neuron position on either plane. For L2 SCs, we further found no relationship between EPSP or IPSP amplitude and neuron position (*Figure 9—figure supplement 1C–D*). In contrast, in L2 PCs the inhibitory component of the membrane potential responses was larger in neurons positioned in medial MEC, close to the border with parasubiculum ($F_{(1,14)}$ = 7.513, p=0.016), and for neurons positioned in ventral MEC ($F_{(1,17)}$ = 4.534, p=0.048). However, there was no relationship between the amplitude of EPSPs in L2 PCs and neuron location. We also did not find evidence for positional dependence in the amplitude of fan cells inputs to interneurons in L1 or L2 (*Figure 9—figure supplement 2*), with the exception of inputs to L2 interneurons which showed a modest decrease with distance from the medial border of the MEC (*Figure 9—figure supplement 2B*). These data suggest that the influence of excitatory inputs is largely independent of neuron location.

In summary, whereas for L2 PCs excitation by fan cell inputs is consistently larger than inhibition, for L2 SCs the E-I ratio is more heterogeneous, with many neurons showing greater inhibition than excitation. This heterogeneity in L2 SC responses is independent of anatomical location and suggests that the net impact of fan cell activation is specific to individual L2 SCs.

## Heterogeneity in responses of stellate cells is maintained during theta frequency stimulation of fan cell inputs

Because during behavioural activation neurons in the superficial LEC show sustained elevation of their firing rates (*Hargreaves et al., 2005*; *Tsao et al., 2018*), we asked whether heterogeneity in the relative amplitude of excitatory and inhibitory responses was maintained during theta frequency stimulation (*Figure 10*). We optically stimulated fan cell axons with trains of light pulses delivered at 10 Hz and measured the corresponding membrane potential responses of L2 SCs and L2 PCs. We find that the striking heterogeneity of L2 SC responses, and the bias of L2 PC responses towards excitation, were both maintained (*Figure 10A–D*). For individual L2 SCs (*Figure 10A*) and L2 PCs (*Figure 10B*) the relative bias towards excitation was similar for responses at the start and end of the 10 Hz train (*Figure 10A–B and D*, *Figure 10—figure supplement 2*). For the L2 SC population there was a trend towards a further increase in heterogeneity (*Figure 10D*), which appeared to be driven by an increase in the amplitude of inhibitory responses (*Figure 10F*) rather than a change in excitatory responses (*Figure 10E*). Notably, these changes were more pronounced with optical stimulation at a frequency of 20 Hz (*Figure 10—figure supplement 1*). Thus, heterogeneity of L2 SC responses and the excitatory bias of L2 PCs appear to be maintained across the physiological range of firing rates likely to be generated by LEC fan cells.

## Discussion

We show that fan cells in the LEC that project to the hippocampus also send axonal collaterals to the superficial MEC. Fan cells directly excite principal cells and their local inhibitory networks, with the balance between direct excitation and feedforward inhibition differing between and within L2 SC and L2 PC populations (*Figure 11A*). We also identify a key role for interneurons in L1 of the MEC in driving slow GABA$_B$ receptor mediated feedforward inhibition of principal cells in L2. Together, our findings establish circuitry by which feature information from the LEC may be integrated with spatial signals in the MEC prior to integration within the dentate gyrus (*Figure 11B*).

### LEC as a relay structure for external inputs to the MEC

Our results suggest a substantial modification to models of entorhinal-hippocampal interactions in which the LEC and MEC are assumed to mediate parallel and independent input streams to the hippocampus. Instead, we show that L2 SCs, L2 PCs, and L3 PCs in the MEC, which together originate major inputs to all hippocampal subfields, each receive substantial inputs from LEC fan cells. These inputs can arise as collaterals of the projections from fan cells to the hippocampal dentate gyrus. Thus, when LEC fan cells are active they will influence the hippocampus directly, through their inputs to the dentate gyrus, and indirectly through their influence on hippocampal projecting neurons in the MEC. It is not yet clear whether this model applies to all fan cells, as our data do not rule out the possibility that fan cells that project to the dentate gyrus and MEC exist alongside fan cells that project to either area alone. Our observation of similar levels of axonal labelling in dentate gyrus and MEC (*Figure 2D*)

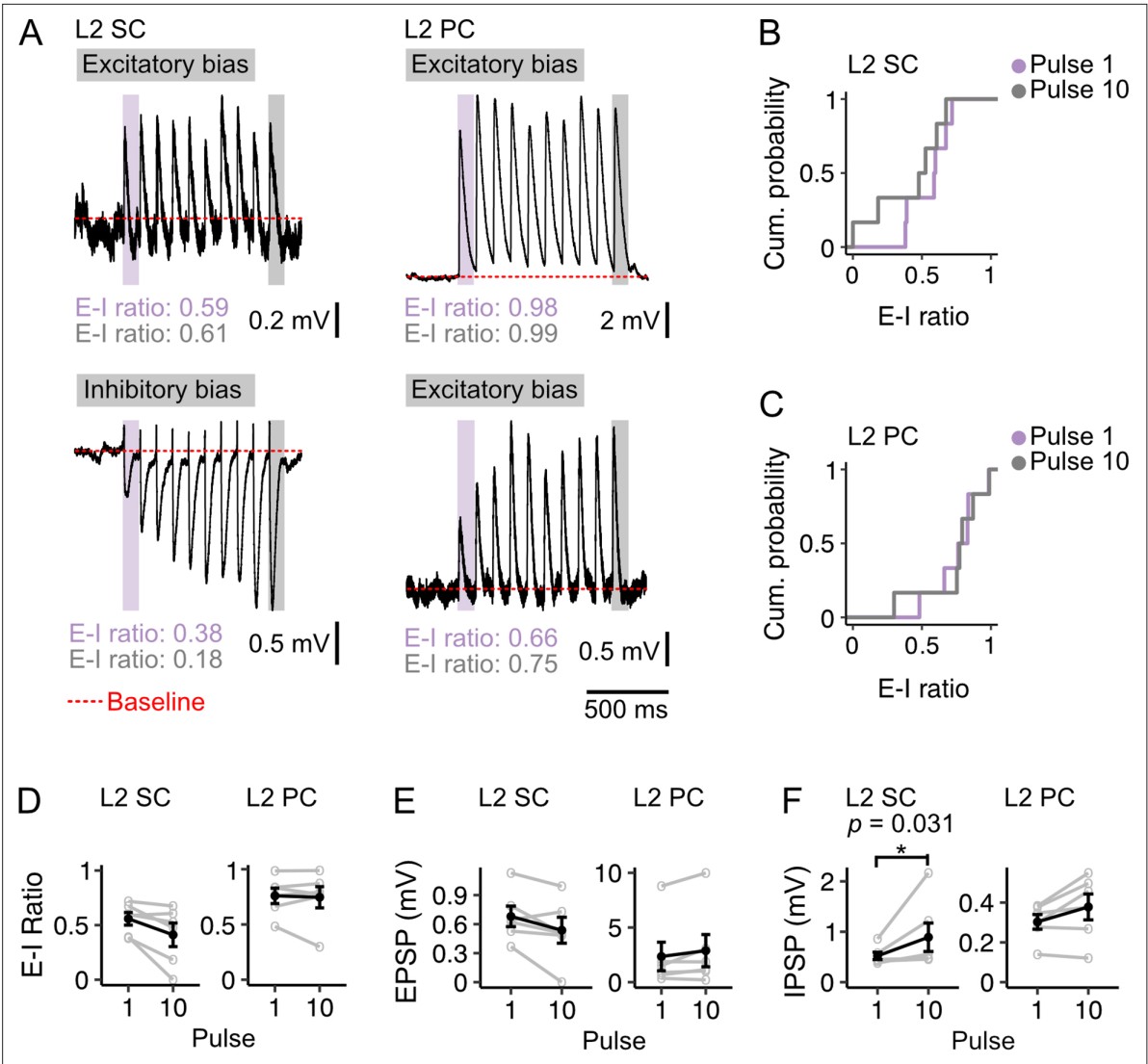

**Figure 10.** Diversity of excitation-inhibition bias is maintained during theta frequency stimulation. (**A**) Example membrane potential responses of layer 2 stellate (L2 SC, left) and pyramidal cells (L2 PC, right) to 10 Hz optical stimulation of fan cell inputs. Examples illustrate relatively high (upper) and low (lower) E-I values for each cell type. Traces are averages of multiple responses (10–30 repetitions). Red dotted line indicates baseline membrane potential (average of the 990 ms window prior to stimulation, with the 10 ms immediately preceding stimulation excluded). (**B-C**) Cumulative probability of the ratios of excitation to inhibition (E-I ratio) of membrane potential responses for pulse 1 (purple) and pulse 10 (grey) for stellate (**B**; Pulse 1: x̄=0.558, IQR = 0.216, Pulse 10: x̄=0.411, IQR = 0.331) and pyramidal cells (**C**; Pulse 1: x̄=0.759, IQR = 0.147, Pulse 10: x̄=0.746, IQR = 0.145). There was more variance in the distribution of E-I ratios for stellate cells at pulse 10 ($D$=0.833, p=0.026, Kolmogorov Smirnov test), but the distributions were not different between cell-types at pulse 1 ($D$=0.667, p=0.143). (**D-F**) Average E-I ratios (**D**), and amplitudes of excitatory (E, EPSP) and inhibitory components (F, IPSP) of responses evoked in stellate (left) and pyramidal (right) cells by the first (1) and last (10) pulse. IPSP amplitudes were larger at pulse 10 for stellate cells ($V$=0, p=0.031, paired samples Wilcoxon test). There were no other differences between pulse 1 and 10 for stellate (EPSP: $V$=19, p=0.094; E-I ratio: $V$=20, p=0.063) or pyramidal cells (EPSP: $V$=4, p=0.219; IPSP: $V$=3, p=0.156; E-I ratio: $V$=10, p=1.00). Black line is the population average and grey lines indicate single neurons. Values for single neurons are calculated as the average of pulses 1 or 10 across all stimulations. Error bars are SEM.

The online version of this article includes the following figure supplement(s) for figure 10:

**Figure supplement 1.** Optogenetic stimulation of fan cell axons at a frequency of 20 Hz.

**Figure supplement 2.** Ratios of excitation to inhibition are similar for potentials evoked by a single stimulus and the first pulse in a train of stimuli.

may suggest that most hippocampal-projecting neurons also project to the MEC, and vice versa, though our data also suggest that the extent of overlap in these populations can vary with different cortical inputs (*Figure 3E and G*). Further evaluation of these possibilities will likely require reconstructions of the axons of individual fan cells.

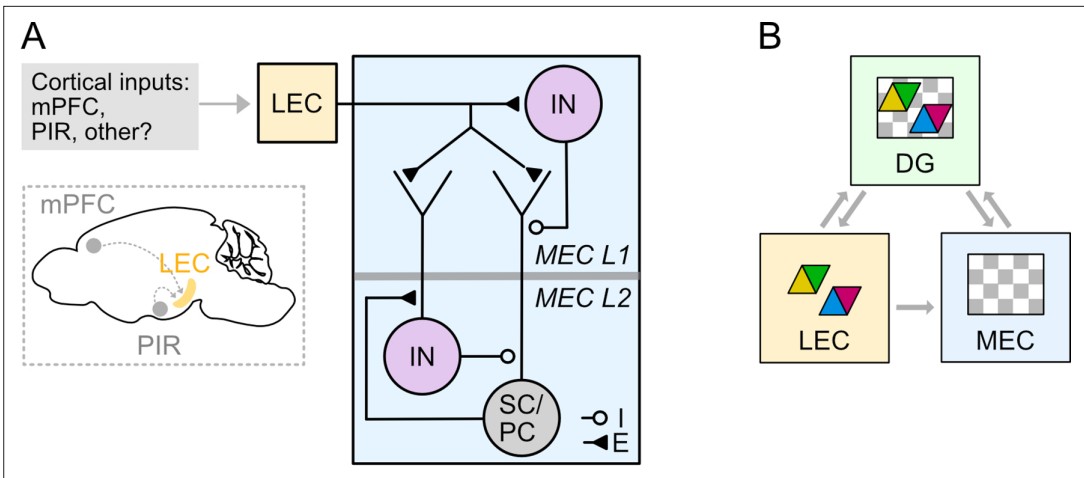

**Figure 11.** Circuitry for higher order and sensory signals to reach medial entorhinal cortex through connectivity with lateral entorhinal cortex. (**A**) Circuit organisation through which neocortical neurons may influence stellate cells (SC), pyramidal cells (PC) and inhibitory interneurons (IN) in the MEC via their projections to LEC. (**B**) A model for entorhinal-hippocampal integration. Grey arrows indicate flow of information about features (coloured triangles) and spatio-temporal context (grey and white chequered triangles) between fan cells in the LEC (yellow) and the hippocampal dentate gyrus (DG, green) and the MEC (blue). Classic models suggest the hippocampus is the main point of convergence of these two types of information. Our data further suggest that feature information could be integrated with spatio-temporal contextual information in the MEC prior to reaching the hippocampus.

Recent analyses of perirhinal and postrhinal inputs to the entorhinal cortex suggest that both structures preferentially target the LEC over the MEC (*Doan et al., 2019*). This result also challenges standard models of entorhinal-hippocampal function, which had assumed that the MEC relays postrhinal input to the hippocampus. However, it leaves open the question of whether and how the MEC receives information from other cortical areas as neurons in the superficial layers of MEC appear to receive few other direct cortical inputs (*Cappaert et al., 2015*; *Doan et al., 2019*). Our findings address this question by showing that LEC fan cells that project to the MEC are downstream from neurons in the prefrontal and piriform cortices, suggesting that cortical signals could reach the MEC via LEC fan cells.

In this revised model, the organisation of inputs to LEC fan cells is likely to be critical for the computations implemented. For example, inputs from multiple cortical structures, in combination with inputs from perirhinal and postrhinal cortex (*Doan et al., 2019*), could converge on single fan cells prior to reaching hippocampus and MEC. Alternatively, different fan cells may receive distinct combinations of input from each structure. Thus, future experiments might distinguish whether fan cells are homogenous, or if there are distinct subpopulations that differ in the origin of their inputs or their preference for targets in the MEC and dentate gyrus.

While it has been recognised that layer 1 of the MEC contains sparsely distributed interneurons, the nature of the input to these neurons and their influence on the activity of other neurons in the MEC has received very little attention. Our data show that layer 1 interneurons are a major target of inputs from LEC fan cells. In contrast to principal cells, responses of layer 1 neurons to activation of fan cell inputs were exclusively excitatory and in many cases were sufficient to drive spike firing. Paired recordings between L1 interneurons and principal cells in L2 demonstrated that postsynaptic responses were a mix of fast and slow inhibitory components. This contrasts with interneurons in L2, which provide exclusively fast inhibitory input to layer 2 principal cells (*Couey et al., 2013*; *Nilssen et al., 2018*; *Pastoll et al., 2013*). Thus, L1 neurons are likely to contribute to feedforward inhibition of L2 SCs and L2 PCs and their activation can account for the slow GABA$_B$ mediated inhibition of these cells following activation of inputs from fan cells. This organisation is similar to neocortical circuits, where activation of interneurons in L1 by long-range cortical inputs drives inhibition of neurons in deeper cortical layers (*Jiang et al., 2015*; *Oláh et al., 2009*; *Schuman et al., 2019*; *Wozny and Williams, 2011*).

What are the implications of this local inhibitory connectivity? In neocortex, interneurons in L1 integrate external sensory information arriving from long-range cortical inputs with internally generated

signals that arise from principal neurons in the deeper layers (*Schuman et al., 2021*). It is possible that circuits encompassing interneurons in L1 have a similar role in the MEC. For example, the deeper layers of MEC contain spatially modulated cells that signal animal position, head direction, and speed (*Sargolini et al., 2006*). If L1 interneurons receive input from these cells, then they would be well placed to integrate these signals with object and event information arriving from the LEC. In this way, they may inhibit hippocampal projecting neurons in superficial layers according to the specific conjunction of these signals.

## The effects of fan cell activation differ between and within target cell types

As well as extending the targets of fan cells in the LEC to include neurons in MEC in addition to the dentate gyrus, our results suggest that the impact of fan cells differs between principal neuron cell types. Responses of granule cells, L2 PCs and L2 SCs to fan cell activation all include an initial EPSP followed by activation of feedforward inhibitory inputs, but the consequences of the inhibition differ. For granule cells, activation of fast feedforward inhibition opposes the excitation provided directly by the fan cell inputs and so reduces the probability of spike firing (*Ewell and Jones, 2010*; *Scharfman, 1991*). In contrast, blocking fast inhibition typically had modest effects on the amplitude of fan cell driven EPSPs in L2 SCs and L2 PCs, but caused a several fold increase in their half-width. Thus, recruitment by fan cells of fast feedforward inhibition in the MEC may limit the time window for summation of direct fast excitation. A further striking difference is that in the MEC activation of fan cells often causes a prominent slow inhibition mediated by GABA$_B$ receptors. The extent to which this slow inhibition manifests differs between cell types: for dentate gyrus granule cells responses appear to be primarily depolarizing; for L2 PCs responses include a mix of depolarization and hyperpolarization but across the population the bias is consistently towards excitation; for L2 SCs responses are most heterogenous spanning the full range from excitation-dominated to inhibition-dominated. These differences suggest distinct functional consequences of fan cell activation for each neuron type. For dentate gyrus granule cells their primary effect may be excitation, with gain control provided by feedforward inhibition (*Elgueta and Bartos, 2019*; *Ewell and Jones, 2010*; *Scharfman, 1991*). For L2 PCs the dominant effect also appears to be excitation, with integration time windows determined by inhibition. For L2 SCs, their effect is much more variable, with many neurons likely to experience net inhibition and others responding in similar ways to L2 PCs.

## Ideas and speculation

Our results, together with recent investigations of peri- and postrhinal inputs to the MEC and LEC (*Doan et al., 2019*), suggest a revised circuit organisation for entorhinal-hippocampal interactions (*Figure 11A*). According to this view, neocortical signals may primarily reach superficial layers of the MEC via fan cells in LEC. Because object representations are well established in LEC (*Deshmukh et al., 2012*; *Deshmukh and Knierim, 2011*; *Keene et al., 2016*; *Tsao et al., 2018*), this pathway could provide signals for generation of object-vector representations in the MEC (*Andersson et al., 2021*; *Høydal et al., 2019*; *Figure 11B*). In principle it could also provide signals for visual representations in the MEC (*Kinkhabwala et al., 2020*), and for encoding of rewarded locations either through modification of grid firing (*Boccara et al., 2019*; *Butler et al., 2019*; *Hardcastle et al., 2017*) or through interruption of ramping activity (*Tennant et al., 2021*). Neurons in LEC also encode representations of time (*Tsao et al., 2018*), therefore our finding of connectivity between fan cells and principal neurons in L3 could suggest a further role for this pathway in supporting temporal representations in CA1 (*Kraus et al., 2013*; *MacDonald et al., 2011*; *Pastalkova et al., 2008*). Nevertheless, this may not be the sole route for neocortical signals to reach MEC. For example, within the hippocampal formation neurons in the pre- and parasubiculum make prominent excitatory connections to neurons in L3 and L2 of the MEC, respectively (*Caballero-Bleda and Witter, 1993*; *Köhler, 1985*; *Swanson and Cowan, 1977*; *Van Groen and Wyss, 1990*). The pre- and parasubiculum may in turn be influenced directly

by inputs from sensory cortical areas (*Köhler, 1985*; *Van Groen and Wyss, 1990*; *Vogt and Miller, 1983*). An important future goal will be to dissociate functions for each of these input pathways. Here, the roles of L1 interneurons in generating slow inhibition, and the striking diversity of responses of L2 SCs to fan cells may be of particular interest, as it raises the possibility of computations that involve active suppression as well as generation of representations in the MEC.

# Materials and methods

**Key resources table**

| Reagent type (species) or resource | Designation | Source or reference | Identifiers | Additional information |
|---|---|---|---|---|
| Strain, strain background (mouse) | Sim1<sup>Cre</sup> | Gen-Sat, MMRC | RRID: MMRRC_034614-UCD | NA |
| Chemical compound, drug | Biocytin | Sigma-Aldrich | B4261 | Biotin-lysine compound used for neuron reconstruction |
| Chemical compound, drug | Gabazine | Hello Bio | HB0901 | GABAA receptor antagonist, diluted to 10 µM |
| Chemical compound, drug | CGP55845 | Hello Bio | HB0960 | GABAB receptor antagonist,diluted to 100 µM |
| Chemical compound, drug | D-AP5 | Hello Bio | HB0225 | NMDA receptor antagonist, diluted to 50 µM |
| Chemical compound, drug | NBQX disodium salt | Hello Bio | HB0442 | AMPA receptor antagonist, diluted to 10 µM |
| Chemical compound, drug | Tetrodotoxin | Hello Bio | HB1034 | Na +channel-blocker, diluted to 500 nM |
| Chemical compound, drug | 4-Aminopyridine (AP) | Hello Bio | HB1073 | Kv channel-blocker, diluted to 200 µM |
| Software, algorithm | R | NA | Version: 3.6.0 | https://www.r-project.org/ |
| Software, algorithm | Matlab | Mathworks | Version: 2013 a | https://www.mathworks.com |
| Software, algorithm | Igor Pro | Wavemetrics | Version: 6.3 | https://www.wavemetrics.com/ |
| Software, algorithm | AxoGraph | AxoGraph | Version 1.7.6 | https://www.axograph.com |
| Software, algorithm | ImageJ | Fiji | NA | https://fiji.sc |
| Other (Stains) | AlexaFluor Streptavidin 647 | Invitrogen | Cat #: S21374 | 1:500 dilution, biotin-binding |
| Other (Stains) | Neurotrace 435/55 (Nissl stain) | Invitrogen | Cat#: N21479 | 1: 500 |
| Other (Stains) | Neurotrace 640/660 (Nissl stain) | Invitrogen | Cat#: N21479 | 1:500 |
| Other (AAV) | AAV1/2-Retro-Ef1a-Cre-WPRE | Addgene | Plasmid #: 51502 | titer: 2.4x10<sup>11</sup> |
| Other (AAV) | AAV1/2-FLEX-GFP | Addgene | Plasmid #: 28304 | titer: 2.44x10<sup>13</sup> |
| Other (AAV) | AAV1/2-FLEX-GFP-2A-Syn-mRuby | Addgene | Plasmid #: 71760 | titer: 2.96x10<sup>12</sup> |
| Other (AAV) | AAV2-FLEX-EF1a-DIO-hChR2(H134R)-EYFP | UNC Vector Core; *Zhang et al., 2006* | | titer: 4x10<sup>12</sup> |
| Other (AAV) | AAV2-Retro-Syn-mCherry | Addgene | Plasmid #: 114472 | titer: 1.9x10<sup>13</sup> |
| Other (AAV) | pENN-AAV1-hSyn-Cre-WPRE | Addgene | Plasmid #: 105553 | titer: 1.8x10<sup>13</sup> |

## Animals

The *Sim1<sup>Cre</sup>* line, which expresses Cre under the control of the Single minded homolog-1 (Sim1) promoter, was generated by Gen-Sat and obtained from MMRRC (strain name: Tg(Sim1cre)KH21Gsat/ Mmucd, RRID:MMRRC_034614-UCD). *Sim1<sup>Cre</sup>* mice were bred to be heterozygous for the Cre transgene by crossing a male mouse carrying the transgene with female C57BL6/J mice (Charles River, UK, strain code: 027). All mice were housed in groups under diurnal light conditions with unrestricted access to food and water. Anatomical experiments used 2- to 6-month-old C57BL6/J and *Sim1<sup>Cre</sup>* mice of both sexes. Electrophysiological experiments used 5–12 week old C57BL6/J and *Sim1<sup>Cre</sup>* mice of

both sexes. All experiments and surgeries were pre-approved by a veterinarian at the University of Edinburgh, and conducted under a project licence administered by the UK Home Office and in accordance with national (Animal [Scientific Procedures] Act, 1986) and international European Communities Council Directive 2010 (2010/63/EU) legislation governing the maintenance of laboratory animals and their use in scientific research. In reporting results of animal research we have aimed to follow the ARRIVE guidelines.

## Stereotaxic injection and viral constructs

For stereotaxic injection of adeno-associated viruses (AAVs), mice were first anaesthetised with Isoflurane in an induction chamber before being transferred to a stereotaxic frame. Mice were administered an analgesic subcutaneously, and an incision was made to expose the skull. For retrograde expression of Cre in neurons that project to the hippocampus, AAV1/2-Retro-Ef1a-Cre-WPRE (Addgene #: 51502, titer: $2.4 \times 10^{11}$ genome copies (GC)/ml, generated in-house) was injected unilaterally into the dorsal dentate gyrus. A craniotomy was made at 2.8 mm posterior to bregma and 1.8 mm lateral to midline. A glass pipette was lowered vertically into the brain and 100–200 nL of virus was injected at a depth of 1.7 mm from the brain surface. We used the following AAVs for injections into lateral entorhinal cortex (LEC): AAV1/2-FLEX-GFP (Addgene #: 28304, titer: $2.44 \times 10^{13}$ GC/ml, generated in-house), AAV1/2-FLEX-GFP-2A-Syn-mRuby (Addgene #: 71760, titer: $2.96 \times 10^{12}$ GC/ml, generated in-house), and AAV2-FLEX-EF1a-DIO-hChR2(H134R)-EYFP (titer: $4 \times 10^{12}$ GC/ml, UNC Vector Core, *Zhang et al., 2006*). Viruses generated in-house were produced using protocols detailed in *Murray et al., 2011*. To target the LEC, a craniotomy was made adjacent to the intersection of the lamboid suture and the ridge of the parietal bone, which was approximately 3.8 mm posterior to bregma and 4.0 lateral to the midline. From these coordinates, the craniotomy was extended 0.8 mm rostrally. At the original coordinates, a glass pipette was lowered from the surface of the brain at an 11° angle until a slight bend in the pipette indicated contact with dura. The pipette was then retracted 0.2 mm and 100–300 nL of virus was injected. This protocol was repeated at a site 0.2 mm rostral to this site. To target ventral LEC, the angle of the pipette was adjusted to 9° and a third injection was delivered at the rostral injection site. For injections into the medial entorhinal cortex (MEC) we used AAV2-Retro-Syn-mCherry (Addgene #: 114472, titer: $1.9 \times 10^{13}$ GC/ml). To target the superficial MEC, a craniotomy was made between the lamboid suture and the transverse sinus at 3.5 mm lateral to midline. A glass pipette was lowered at a 10° angle until a slight bend in the pipette indicated contact with dura. The pipette was retracted 0.2 mm and 100–300 nL of virus was injected. To target the dorsoventral extent of MEC, the injection protocol was repeated with the pipette angled at 8°. For anterograde expression of Cre in neurons that receive projections from cortex, pENN-AAV1-hSyn-Cre-WPRE (Addgene #: 105553, titer: $1.8 \times 10^{13}$ GC/ml) was injected into the piriform cortex or medial prefrontal cortex. To target piriform cortex, a craniotomy was made at 0.6 mm posterior to bregma and 3.3 mm lateral to the midline, a glass pipette was lowered vertically into the brain to a depth of 4 mm from dura, and 200 nL of virus was injected. To target medial prefrontal cortex, a craniotomy was made at 1 mm rostral to bregma and 0.35 mm lateral to midline, a glass pipette was lowered into the brain angled at 15° to a depth 1.34 mm from dura, and 200 nL of virus was injected. For all injections, the pipette was slowly retracted after a stationary period of four minutes. After injections, the incision was closed with tissue glue (Vetbond) or sutures. Mice were administered an oral analgesic prepared in flavoured jelly immediately after surgery.

## Immunohistochemistry

For anatomical experiments, animals were sacrificed 3–8 weeks after viral injections. Mice were administered a lethal dose of sodium pentobarbital and transcardially perfused with cold phosphate buffered saline (PBS) followed by cold paraformaldehyde (PFA, 4%). For immunohistochemistry in slices, brains were fixed for minimum for 24 hr in PFA at 4 °C, washed with PBS, and transferred to a 30% sucrose solution prepared in PBS for a minimum for 48 hr at 4 °C. Brains were then sectioned horizontally at 50–60 μm on a freezing microtome. To counterstain neurons, slices were incubated in a Neurotrace 640/660 (Invitrogen, Cat # N21483, 1:500) solution prepared in 0.3% PBS-T (Triton) for 24 hr at room temperature. Slices were washed with 0.3% PBS-T 3 x for 20 min and then mounted and cover-slipped with Mowiol. Mounted sections were stored at 4 °C.

## Slice electrophysiology

Horizontal brain slices were prepared from 5- to 12-week-old *Sim1*[Cre] or wild-type mice. Where experiments relied on expression of AAV, animals were sacrificed 2–4 weeks after viral injections. Mice were sacrificed by cervical location and decapitated. The brains were rapidly removed and submerged in cold cutting artificial cerebrospinal fluid (ACSF) at 4–8°C. The cutting ACSF was composed of the following (in mM): NaCl 86, $NaH_2PO_4$ 1.2, KCl 2.5, $NaHCO_3$ 25, Glucose 25, Sucrose 50, $CaCl_2$ 0.5, $MgCl_2$ 7. The dorsal surface of the brain was glued to a block submerged in cold cutting ACSF and 400–450 µm thick horizontal slices were cut using a vibratome. Slices were transferred to standard ACSF at 37 °C for a minimum of 15 min, then incubated at room temperature for a minimum of one hour. Standard ACSF consisted of the following (in mM): NaCl 124, $NaH_2PO_4$ 1.2, KCl 2.5, $NaHCO_3$ 25, Glucose 25, $CaCl_2$ 2, $MgCl_2$ 1. For recordings, slices were transferred to a submerged chamber and maintained in standard ACSF at 35–37°C. Whole-cell patch clamp recordings were made from neurons in MEC using borosilicate electrodes with a resistance of 3–8 MΩ. Electrodes with resistances in the higher end of this range (6–8 MΩ) were found to be optimal for recording from interneurons in layer 1 of the MEC. Electrodes were filled with an intracellular solution comprised of the following (in mM): K gluconate 130, KCl 10, HEPES 10, $MgCl_2$ 2, EGTA 0.1, $Na_2ATP$ 4, $Na_2GTP$ 0.3, phosphocreatine 10, and 0.5% biocytin (w/v). Recordings were made in current-clamp mode from cells with a resting membrane potential ≤ –50 mV and series resistance ≤ 50 MΩ with appropriate bridge-balance and pipette capacitance neutralisations applied.

## Recording protocols

A series of protocols were used to characterise the electrophysiological properties of each cell recorded in MEC. Sub-threshold membrane properties were measured by examining membrane potential responses to injection of current in hyperpolarizing and depolarizing steps (−160–160 pA in 80 pA increments, or –80–80 pA in 40 pA increments, each 3 s duration), and to injection of an oscillatory current with a linearly varying frequency (ZAP protocol) (*Nolan et al., 2007*). Suprathreshold properties were estimated from responses to depolarizing current ramps (50 pA/s, 3 s). Connectivity between cells in layer 1 and layer 2 of MEC was established by performing simultaneous patch-clamp recordings and measuring the changes in membrane potential of one cell during injection of current to evoke an action potential (1–2 nA, 3ms) in the other cell. Responses to optogenetic activation of LEC fiber inputs to MEC were evaluated by stimulation with 470 nm wavelength light for 3ms at 22.4 $mW/mm^2$. The majority of neurons had subthreshold membrane potential responses to stimulation at this intensity, but for neurons in which action potentials were evoked (n=15) the threshold intensity required to elicit spiking responses was established and stimulation protocols were repeated at an intensity just below this threshold. To test roles of glutamate and GABA receptors, a baseline series of responses to light stimulation were recorded, glutamate or GABA receptor antagonists were bath-applied, and then responses to light stimulation were re-evaluated using the same stimulation protocols. Glutamatergic transmission was blocked using antagonists for AMPA (NBQX, 10 µM) and NMDA receptors (APV, 50 µM). GABAergic transmission was blocked using antagonists for $GABA_A$ (Gabazine, 10 µM) and $GABA_B$ receptors (CGP 55485, 100 µM). To isolate monosynaptic connections, the response to light stimulation was re-evaluated after bath application of the $Na^+$ channel-blocker tetrodotoxin (500 nM) and Kv channel-blocker 4-AP (200 µM). Upon completion of investigatory protocols, diffusion of biocytin into the cell was encouraged by injecting large depolarizing currents into the cell (15x4 nA, 100ms steps, 1 Hz). Each cell was left with the electrode attached for up to 90 min before being transferred to PFA (4%). Slices were stored at 4 °C for a minimum of 24 hr before histological processing.

## Recovery of neuronal morphology

Fixed slices containing biocytin-filled neurons were washed with PBS 4 x for 10 min and transferred to a solution containing AlexaFluor Streptavidin 647 (Invitrogen, Cat # S21374, 1:500) and Neurotrace 435/55 (Invitrogen, Cat # N21479, 1:500) in 0.3% PBS-T for 24–48 hr at room temperature. Slices were washed with PBS-T 4 x for 20 min and then mounted and cover-slipped with Mowiol. Mounted sections were stored at 4 °C.

## Microscopy

Epifluorescent images were acquired using a Zeiss Axioscan slide scanner with ZenPro software. Confocal images were acquired using a Zeiss LSM800 confocal microscope and ZenPro software. To confirm the location of virus expression in the LEC or MEC, images were acquired at a ×10 magnification on the epifluorescent microscope or using a 10 x or 20 x objective on the confocal microscope. Images used for quantification of synaptic puncta in the MEC or establishing the morphology and position of biocytin-filled neurons were acquired on the confocal microscope. For quantification of synaptic puncta, z-stacks were acquired of regions of interest (ROI) at 1 μM steps using a 40 x oil-dipped objective. For imaging of neurons after slice electrophysiology, z-stacks were acquired of biocytin-filled cells at 1–2 μM steps using a 20 x objective. Where possible, different cell types were distinguished by visual comparison of the shape of the soma and arrangement of dendrites to published morphological descriptions.

## Quantification of immunohistochemistry

For anatomical experiments, the location of virus expression was determined by referencing an atlas of the mouse brain (*Paxinos and Franklin, 2003*). Puncta density and fluorescence intensity of fan cell axons was quantified in ImageJ (http://fiji.sc) using the 3D Objects Counter plug-in (https://imagej.net/3D_Objects_Counter). Threshold values were set for each ROI to subtract background fluorescence. For quantification across layers, a rectangular extract (150 μm wide) was taken from the ROI for each layer of MEC. For quantification across the mediolateral axis of MEC, six sequential identical rectangular extracts (150 μm wide) were taken from the ROI to span the mediolateral extent of layer 1, starting from the border of MEC with the parasubiculum. Within each extract, volumetric puncta density was calculated as puncta count/tissue volume (mm$^3$). To compare the fluorescence intensity of fan cell axons labelled in the dentate gyrus (DG) and MEC, ROIs (100 x 50 μm) were taken from the outer molecular layer (oml) of DG and layer 1 of MEC. Pixel intensity was calculated as a gray value from a possible range of 1–255, where 1 corresponds to black and 255 corresponds to white. Mean pixel intensity was calculated for each ROI, and the mean pixel intensity of a baseline ROI of the same size, taken from a region of perirhinal cortex with no labelled axons, was subtracted from this value. For comparison of fluorescence intensity with location, the RGB profiler plug-in (https://imagej.net/plugins/rgb-profiler) was used to extract mean pixel intensities across a 400 x 100 μm (MEC) or 200 x 100 μm (DG) window to produce expression plots showing changes in intensity across MEC or DG layers. ROIs were oriented so that layers were perpendicular to distance. Values were normalised to the minimum and maximum fluorescence values using the formula: (intensity value - minimum intensity value) / (maximum intensity value - minimum intensity value). The slice with the brightest fluorescence from each animal was chosen for these analyses.

## Analysis of electrophysiological data

Neurons were classified into different types by their electrophysiological profiles and morphology. Interneurons in layer 2 of the MEC were classified as either low-threshold spiking or fast-spiking using criteria described in *Gonzalez-Sulser et al., 2014*. Electrophysiological data were analyzed with AxoGraph (https://axographx.com), IGOR Pro (Wavemetrics, USA) using Neuromatic (http://www.neuromatic.thinkrandom.com), and customised MATLAB scripts. Input resistance, time constant and time-dependent inward rectification ('sag') were measured from the membrane potential response to hyperpolarizing current steps (–80 or –40 pA). Input resistance was estimated by dividing the steady-state voltage change from the resting membrane potential by the amplitude of the injected current. Time constant was estimated as the time taken for the change in voltage to reach ~63% of its steady-state value. Sag was measured as the ratio between the maximum decrease in voltage and the steady-state decrease in voltage. Rheobase was measured as the minimum amplitude of depolarizing current which elicited an action potential response. Action potential duration was measured from the action potential threshold, which was defined as the point at which the first derivative of the membrane potential exceeded 1 mV ms$^{-1}$. Action potential amplitude was measured as the change in voltage between the action potential threshold and peak. To determine the resonant frequency of the cell, membrane impedance was first calculated by dividing the Fourier transform of the membrane voltage response by the Fourier transform of the input current from the ZAP protocol, which was then converted into magnitude and phase components. The resonant frequency was defined as the

input frequency which corresponded to the peak impedance magnitude. To quantify the response of MEC neurons to optogenetic stimulation of inputs from LEC, the change in amplitude was measured between baseline resting membrane potential (average of 490ms window before stimulation, excluding the 10 ms immediately prior to stimulation) and the membrane response to stimulation (average of 1ms window after stimulation at peak depolarisation and hyperpolarisation). For experiments where trains of stimuli were delivered, the baseline window was calculated as the average of a 990 ms window before stimulation (excluding the 10 ms immediately prior to stimulation) and the change in amplitude for each pulse was measured as the peak depolarisation and hyperpolarisation in the window (100 ms for 10 Hz or 50 ms for 20 Hz stimulation) between pulses. The latency and half-width were usually measured for postsynaptic potentials with amplitudes ≥ 1 mV. For half-width measurements of excitatory potentials from experiments with GABA receptor antagonists (*Figure 6*) or half-width measurements of inhibitory potentials evoked by stimulation of layer 1 interneurons in paired recording experiments (*Figure 8*), half-width values were calculated from the single average of all pulses to optimise measurement of smaller potentials. The latency of postsynaptic potentials was measured as the time taken from stimulus onset to reach a10 % deviation from baseline membrane potential. The ratios of excitation to inhibition in postsynaptic potentials evoked after stimulation of fan cell axons were calculated using the following formula: ratio = mean peak amplitude of excitatory component/(mean peak amplitude of excitatory component-absolute value of the mean peak amplitude of the inhibitory component).

## Establishing neuron position in MEC

For slice electrophysiology experiments, the depth of the slice from bregma and the borders of the LEC and the MEC were determined by referencing an atlas of the mouse brain (*Paxinos and Franklin, 2003*). For each biocytin-filled cell in the MEC, the encompassing layer was determined by examining the soma position in relation to laminar delineation with counterstaining using Neurotrace. The distance of the soma from the medial MEC border with the parasubiculum was measured in ImageJ using a built-in tool calibrated to the scale of the image. Where tissue damage or insufficient reconstruction of the neuron prevented precise localisation of the neuron to a dorsoventral or mediolateral position, the neuron was excluded from analyses of neuron position in relation to membrane response properties. In slices with expression of AAV in the LEC, the region of virus expression was examined in relation to the border between LEC and MEC. Data was discarded from all animals with any expression of virus in MEC or apparent labelling of MEC axons in their established termination zone in the middle molecular layer of the dentate gyrus.

## Statistical analyses

Statistical analyses and plots were generated using R (https://www.r-project.org/). Non-parametric statistics were used in all analyses because our datasets contained non-normal distributions, small samples sizes, and unequal group sizes. To compare puncta density across layers or the mediolateral axis of the MEC, we used a repeated measures Friedman test with layer or distance of the cell body from the MEC border with parasubiculum as factors. The effect size of the result was quantified with Kendall's *W*. A Mann-Whitney U test was used to compare the fluorescence intensity of fan cell axons in the dentate gyrus and MEC. Posthoc comparisons for repeated measures Friedman tests were performed using pairwise Wilcoxon tests with a Bonferroni adjustment for multiple comparisons. To compare membrane response amplitudes or latencies across principal cell types, we performed Kruskall-Wallis tests with cell type as a between-subjects factor. Posthoc pairwise comparisons for Kruskal-Wallis tests were performed using a Dunn test with a Bonferroni correction for multiple comparisons. Analysis of measurements of response amplitudes and width after application of receptor antagonists was performed using a Friedman test with receptor antagonist as a within-subjects factor and data grouped by either cell or stimulus sweep (n sweeps: 30–50). A Mann-Whitney U test was used to compare postsynaptic response amplitudes and latencies across layer 1 and layer 2 interneurons and to compare ratios of excitation to inhibition across layer 2 stellate and pyramidal cells. Distributions of values were compared between cell types using Kolgmorov-Smirnov

tests. The relationships between the amplitude of excitatory and inhibitory components, resting membrane potentials and membrane response characteristics, or neuron location and membrane response characteristics were quantified using simple linear regression models with the equation $y = \beta_0 + \beta_1 x + \epsilon$, where $\beta_0$ is the estimate of the intercept, $\beta_1$ is the estimate of the effect of $x$ on $y$, and $\epsilon$ is a random variable/error. Model fit was quantified using an $F$-statistic. Amplitude measurements between single pulse data, and data from the first and last sweep of 10 Hz or 20 Hz stimulation were compared using paired Wilcoxon tests.

## Data availability statement

Data will be made available at https://datashare.ed.ac.uk/handle/10283/777. Source data and code is available at https://github.com/MattNolanLab/lec_to_mec; *Nolan Lab, 2022*.

## Acknowledgements

This work was supported by grants from the Wellcome Trust (200855/Z/16/Z) to MFN and the BBSRC (BB/V010107/1) to MFN and BV. The authors thank Innes Jarmson for the generation of adeno-associated viruses.

## Additional information

### Funding

| Funder | Grant reference number | Author |
| --- | --- | --- |
| Wellcome Trust | 200855/Z/16/Z | Matthew F Nolan |
| Biotechnology and Biological Sciences Research Council | BB/V010107/1 | Brianna Vandrey Matthew F Nolan |

The funders had no role in study design, data collection and interpretation, or the decision to submit the work for publication. For the purpose of Open Access, the authors have applied a CC BY public copyright license to any Author Accepted Manuscript version arising from this submission.

### Author contributions

Brianna Vandrey, Conceptualization, Formal analysis, Funding acquisition, Investigation, Visualization, Methodology, Writing – original draft, Writing – review and editing; Jack Armstrong, Formal analysis, Investigation, Visualization, Writing – review and editing; Christina M Brown, Investigation, Writing – review and editing; Derek LF Garden, Investigation, Methodology, Writing – review and editing; Matthew F Nolan, Conceptualization, Supervision, Funding acquisition, Writing – original draft, Project administration, Writing – review and editing

### Author ORCIDs

Derek LF Garden http://orcid.org/0000-0003-3336-3791
Matthew F Nolan http://orcid.org/0000-0003-1062-6501

### Ethics

All experiments and surgeries were pre-approved by a veterinarian at the University of Edinburgh, and conducted under a project licence administered by the UK Home Office and in accordance with national (Animal [Scientific Procedures] Act, 1986) and international European Communities Council Directive 2010 (2010/63/EU) legislation governing the maintenance of laboratory animals and their use in scientific research. In reporting results of animal research we have aimed to follow the ARRIVE guidelines.

### Decision letter and Author response

Decision letter https://doi.org/10.7554/eLife.83008.sa1
Author response https://doi.org/10.7554/eLife.83008.sa2

## Additional files

### Supplementary files
• MDAR checklist

### Data availability
Data is available at: https://doi.org/10.7488/ds/3787 Source data and code is available at https://github.com/MattNolanLab/lec_to_mec, (copy archived at swh:1:rev:7133694bfb6acdb42888c11ea73e1e8b7e8b6939).

The following datasets were generated:

| Author(s) | Year | Dataset title | Dataset URL | Database and Identifier |
|---|---|---|---|---|
| Vandrey B, Armstrong J, Brown CM, Garden DLF, Nolan MF | 2022 | Fan cells in lateral entorhinal cortex directly influence medial entorhinal cortex through synaptic connections in layer 1 | https://doi.org/10.7488/ds/3787 | Edinburgh DataShare, 10.7488/ds/3787 |
| Nolan M | 2022 | MattNolanLab/lec_to_mec: Publication code | https://doi.org/10.5281/zenodo.7494950 | Zenodo, 10.5281/zenodo.7494950 |

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
