## [Editor Report]

This is an important manuscript that convincingly reveals a novel pathway by which the lateral entorhinal cortex directly projects to the medial entorhinal cortex. This work thus revises the traditional models that envision lateral and medial entorhinal cortex as providing independent inputs to the hippocampus. Instead, the work points to these cortical regions as participating in the combination of spatial information with sensory and other high-order signals even before routing information to the hippocampus for memory formation.

---

## [Decision Letter]

**Decision letter after peer review:**

Thank you for submitting your article "Fan cells in lateral entorhinal cortex directly influence medial entorhinal cortex through synaptic connections in layer 1" for consideration by *eLife*. Your article has been reviewed by 3 peer reviewers, and the evaluation has been overseen by a Reviewing Editor and John Huguenard as the Senior Editor. The following individuals involved in the review of your submission have agreed to reveal their identity: Kei Igarashi (Reviewer #1); Mark Brandon (Reviewer #2).

Essential revisions:

1) Reviewers 1 and 2 have a number of comments regarding citations and text changes that should be made. Reviewer 3 also has some suggestions regarding combining figures that should be considered but are not necessary for the revision.

2) Reviewer 2 suggests quantifying the proportion of fan cells that project to both MEC and DG. If this is possible to do, I would recommend adding it, as it would add significant relevance to this report.

3) Reviewer 3 raises a few points that should be addressed. First, to show population data and statistical analyses for the points in Figures 2 and 3, rather than just example images. Second, discussion regarding why NBQX does not completely block the ESPSs and disynaptic IPSPs (see more detailed comment below). Third, discussion regarding the chloride reversal potential (see more detailed comment below).

*Reviewer #1 (Recommendations for the authors):*

1. Page 2 Line 5. The reference on the parallel streams from the LEC and MEC needs to include several established reviews by Rebecca Burwell, Menno Witter, and Jim Knierim.

2. Page 2 Line 10. "neurons in the LEC encode local features of the environment"

Please include Igarashi et al., 2014 and Lee et al 2021 here. As these works showed the encoding of olfactory memory by LEC neurons, the sentence might be like this: "neurons in the LEC encode local features and items near animals generated from tactile and olfactory inputs."

3. Page 3 Line 1. "fan cells in LEC have been shown to contribute to the processing of olfactory information (Lee et al 2021; Leitner et al., 2016)"

Lee et al paper showed encoding of olfactory memory by fan cells whereas Leitner paper showed only responses of fan cells to olfactory stimuli in anaesthetized animals. The sentence should be e.g. "fan cells in LEC have been shown to contribute to the processing (Leitner et al., 2016) and memory encoding (Lee et al 2021) of olfactory information."

4. Page 3 Line 2. The Fernandez-Ruiz paper inhibited the LEC using optogenetic stimulation of interneurons in the LEC, but did not manipulate fan cells. If the focus of the sentence is fan cells, this reference needs to be removed.

5. Page 3 Line 17. "For example, ..."

I had difficulty understanding this sentence - please elaborate.

*Reviewer #2 (Recommendations for the authors):*

Introduction (second paragraph)

"These projections primarily originate from reelin expressing neurons, which in MEC are LEC are referred to respectively as stellate cells (L2 SCs) and fan cells (L2 FCs) on the basis of their distinct dendritic morphology". Are → and.

Interneurons in L1 of MEC mediate slow components of inhibitory input to layer 2 principal neurons

"The EPSPs were abolished after application of NBQX and AP-5 (Figure 7E), indicating that they are glutamatergic, and were maintained in the presence of TTX and 4-AP, indicating that they result from direct monosynaptic activation (Figure 7F)" (end of the second paragraph) EPSPs in L1/L2 INs are abolished in the presence of TTX but recovered after application of 4-AP.

Comments

It would be nice to have an idea of the proportion of fan cells that project to both MEC and DG. This would add significant relevance to this report, especially if the point of this study is to revise the classic entorhinal-hippocampal circuit. This point is discussed at the beginning of the Discussion section. However, elucidating this question is in my opinion quite important to speculate about the relevance of this new route of communication between the entorhinal cortex and the hippocampus.

Since the authors also observed that a fair proportion of L2 SC in the LEC project to L3 PC in the MEC (that project to CA1) and that L2 SCs make a rich connection with L1 IN. Thus, I would also discuss the implication of this result regarding the classic view of the temporammonic pathway function. As the authors speculate that this new route might be the first stage of integrating "what" and "where", through L3 PC to CA1, it might also have important implications in the "when" information.

*Reviewer #3 (Recommendations for the authors):*

I feel that there are more figures than necessary to get the most straightforward points the authors are making across. Figures 1-3 seem like they could be combined, as they are all making points about the anatomy which aren't overly complex individually. Maybe figures 7 and 8 and 9 and 10 could also be combined, as they are thematically similar.

I think it's important to show population data and statistical analyses for the points made in Figures2 and 3 rather than just example images.

Two potential issues of concern with the physiology:

1. Why does NBQX not completely block the EPSPs and disynaptic IPSPs? It would be highly unusual for these to be mediated by NMDARs in the absence of AMPARs at hyperpolarized voltages and normal extracellular divalent concentrations. The authors do not mention anything about this. An NMDAR-mediated form of transmission would be particularly interesting in this circuit and is worth further discussion and study.

2. The authors are rightly interested in the effects of inhibition in this circuit but don't seem to discuss the chloride reversal potential, which may be critically important for mediating the effects they have observed. They should specifically deal with this point.

---

## [Author Response]

Essential revisions:1) Reviewers 1 and 2 have a number of comments regarding citations and text changes that should be made. Reviewer 3 also has some suggestions regarding combining figures that should be considered but are not necessary for the revision.

We have revised the manuscript to address comments made by the reviewers. The changes are detailed in the point by point response below. We note that we carefully considered the suggestion to combine the figures, but concluded that the presentation is clearer if the figures are separate.

2) Reviewer 2 suggests quantifying the proportion of fan cells that project to both MEC and DG. If this is possible to do, I would recommend adding it, as it would add significant relevance to this report.

We now report estimates that give a lower bound on this proportion (~ 22%) and consider additional evidence suggesting the proportion may be much higher. In the discussion we note that definitive numbers will require future studies that fully reconstruct axons of many fan cells.

3) Reviewer 3 raises a few points that should be addressed. First, to show population data and statistical analyses for the points in Figures 2 and 3, rather than just example images. Second, discussion regarding why NBQX does not completely block the ESPSs and disynaptic IPSPs (see more detailed comment below). Third, discussion regarding the chloride reversal potential (see more detailed comment below).

We have added the requested analyses and have addressed below the comments regarding the NMDA component of EPSPs and the Cl^-^ reversal potential.

In addition to these changes we have made further changes to the manuscript to improve the presentation of the work. We have also added additional details to improve the presentation of statistical analyses associated with Figure 6 (included as Figure 6 Supplements 2 and 3).

Reviewer #1 (Recommendations for the authors):1. Page 2 Line 5. The reference on the parallel streams from the LEC and MEC needs to include several established reviews by Rebecca Burwell, Menno Witter, and Jim Knierim.

These references have been added (p 2, lines 37-38).

2. Page 2 Line 10. "neurons in the LEC encode local features of the environment"Please include Igarashi et al., 2014 and Lee et al 2021 here. As these works showed the encoding of olfactory memory by LEC neurons, the sentence might be like this: "neurons in the LEC encode local features and items near animals generated from tactile and olfactory inputs."

We have added the additional citations (p 2, lines 42-45 ). We have not changed the text as the original sentence is more general (because items are features) and the specific sensory modalities are not addressed in the experiments we report here.

3. Page 3 Line 1. "fan cells in LEC have been shown to contribute to the processing of olfactory information (Lee et al 2021; Leitner et al., 2016)"Lee et al paper showed encoding of olfactory memory by fan cells whereas Leitner paper showed only responses of fan cells to olfactory stimuli in anaesthetized animals. The sentence should be e.g. "fan cells in LEC have been shown to contribute to the processing (Leitner et al., 2016) and memory encoding (Lee et al 2021) of olfactory information."

We have modified the sentence following the suggestion (p 3, lines 67-68).

4. Page 3 Line 2. The Fernandez-Ruiz paper inhibited the LEC using optogenetic stimulation of interneurons in the LEC, but did not manipulate fan cells. If the focus of the sentence is fan cells, this reference needs to be removed.

It is correct that the optogenetic stimulation in this study is of interneurons and not fan cells. However, collectively the evidence in this paper argues that the behavioural deficits result from disruption of communication between the MEC/LEC and dentate gyrus/CA3. Because fan cells are the source of input from the LEC to the dentate gyrus/CA3, the evidence in the study therefore points to a role of fan cells in learning of object locations. Given this we have kept the sentence as it is. Hopefully the question raised and our response here will be helpful to readers interested in a more nuanced interpretation.

5. Page 3 Line 17. "For example, ..."I had difficulty understanding this sentence - please elaborate.

We have modified this sentence to try to make it clearer (p 3, lines 89-91).

Reviewer #2 (Recommendations for the authors):Introduction (second paragraph)"These projections primarily originate from reelin expressing neurons, which in MEC are LEC are referred to respectively as stellate cells (L2 SCs) and fan cells (L2 FCs) on the basis of their distinct dendritic morphology". Are → and.

We have corrected this (p 2, line 55).

Interneurons in L1 of MEC mediate slow components of inhibitory input to layer 2 principal neurons"The EPSPs were abolished after application of NBQX and AP-5 (Figure 7E), indicating that they are glutamatergic, and were maintained in the presence of TTX and 4-AP, indicating that they result from direct monosynaptic activation (Figure 7F)" (end of the second paragraph) EPSPs in L1/L2 INs are abolished in the presence of TTX but recovered after application of 4-AP.

This sentence has been corrected (p 9, lines 294-296).

CommentsIt would be nice to have an idea of the proportion of fan cells that project to both MEC and DG. This would add significant relevance to this report, especially if the point of this study is to revise the classic entorhinal-hippocampal circuit. This point is discussed at the beginning of the Discussion section. However, elucidating this question is in my opinion quite important to speculate about the relevance of this new route of communication between the entorhinal cortex and the hippocampus.

Our results address this in two ways:

First, the retrograde labelling experiment presented in Figure 1A establishes a lower bound (p 4, line 123). These data show that ~ 22 % of fan cells are labelled in the LEC following injection of retrograde tracers into the MEC. Because the efficiency of labelling is unknown, and is likely to be incomplete, it is probable that the actual proportion is much higher.

Second, we have compared fluorescence intensity of axons in dentate gyrus and MEC after labelling fan cells with different strategies. When fan cells were labelled using the Sim1:Cre mouse line (Figure 1B) or based on their projection to the hippocampus (Figure 2A), the intensity of labelling appears similar in the MEC and DG (Figure 1B, Figure 2D, E). This would be expected if the average efficacy of projections to each area is similar. When fan cells were labelled by their input from the prefrontal cortex, their axonal fluorescence in the MEC and DG was similar (Figure 3G), although when they were labelled by their input from the piriform cortex, their axonal fluorescence appeared greater in the DG compared to MEC (Figure 3D), hinting at a degree of heterogeneity in the organisation of the fan cell subpopulations. These analyses are referenced in the main text (p 5, lines 151-153; p 12, lines 410-412) and described in the Methods (p 21, lines 682-695; p 23, lines 761-762).

In considering this point it is important also to recognise that our electrophysiological experiments clearly demonstrate a substantial functional impact of the pathway. This holds whether all DG projecting fan cells project to MEC, or whether only a subset do. In either case our data argue for functional importance of the pathway, although in each scenario the computational role would be expected to differ.

In summary, our analyses support the interpretation that many (and perhaps all) fan cells project to both the DG and the MEC, and clearly show the substantial functional impact of this projection. While delineating variability between individual fan cells in their connectivity is beyond the scope of the experiments described here, we agree that this will be an important question to address in the future. We therefore note in the Discussion that further answers to the Reviewer’s question will likely require reconstructions of the axons from many individual fan cells (p 12, lines 411-412).

Since the authors also observed that a fair proportion of L2 SC in the LEC project to L3 PC in the MEC (that project to CA1) and that L2 SCs make a rich connection with L1 IN. Thus, I would also discuss the implication of this result regarding the classic view of the temporammonic pathway function. As the authors speculate that this new route might be the first stage of integrating "what" and "where", through L3 PC to CA1, it might also have important implications in the "when" information.

This is an important point which we now consider in the Ideas and Speculation part of the discussion (p 15, lines 493-497).

Reviewer #3 (Recommendations for the authors):I feel that there are more figures than necessary to get the most straightforward points the authors are making across. Figures 1-3 seem like they could be combined, as they are all making points about the anatomy which aren't overly complex individually. Maybe figures 7 and 8 and 9 and 10 could also be combined, as they are thematically similar.

We have carefully considered this suggestion. Though there is thematic overlap in some instances, the question addressed by each figure is distinct. We therefore feel that condensing the figures would reduce clarity. The inevitable reduction in size of the panels would also make it harder for a reader to assess the data. Given that *eLife* does not have constraints on space we have therefore kept the number of figures the same as in the original submission.

I think it's important to show population data and statistical analyses for the points made in Figures2 and 3 rather than just example images.

We have now added measurements of axonal fluorescence intensity in dentate gyrus and MEC to Figures 2 and 3. This data shows an increase in layer 1 relative to the baseline fluorescence in both areas for all experiments across all animals. We refer to these data in the Results section (p 5, lines 151-153; p 12, lines 410-412) and described the additional analyses in the methods section (p 21, lines 682-695; p 23, lines 761-762).

Two potential issues of concern with the physiology:1. Why does NBQX not completely block the EPSPs and disynaptic IPSPs? It would be highly unusual for these to be mediated by NMDARs in the absence of AMPARs at hyperpolarized voltages and normal extracellular divalent concentrations. The authors do not mention anything about this. An NMDAR-mediated form of transmission would be particularly interesting in this circuit and is worth further discussion and study.

In our view the most likely explanation, and also the most mundane, is that the block of NMDA receptors by physiological Mg^2+^ is incomplete at membrane potentials between -60 mV and -70 mV. This is well established (e.g. see Wrighton et al., J. Physiol., 2008) although perhaps often overlooked. Two more exotic possible explanations also come to mind. First, it may be that the remaining response is mediated by NMDA receptors assembled from the less Mg^2+^ sensitive GluN2C and GluN2D subunits (e.g. see Wyllie et al. Neuropharmacology, 2013). This seems unlikely given their expression level is low in the mature MEC. Second, it may be that the synaptic locations are depolarised relative to the recording site at the soma. Because both ideas are speculative at this point and as they are not central to the core hypotheses we aim to test, we have not addressed them further in the text. We hope the comment here will be of use to an interested reader.

2. The authors are rightly interested in the effects of inhibition in this circuit but don't seem to discuss the chloride reversal potential, which may be critically important for mediating the effects they have observed. They should specifically deal with this point.

We assume here that the reviewer is referring to whether differences in the Cl^-^ reversal potential could contribute to heterogeneity of the responses of L2SCs. Because in our experiments the recorded neurons are dialysed with the pipette solution we expect the chloride reversal potential to be similar between neurons. Variability in the E-I ratio could result from differences in the baseline membrane potential relative to the Cl^-^ reversal potential. However, we find that in L2 SCs the E-I ratio is independent of the baseline membrane potential (discussed on p 10, para 2 and shown in Figure 9, Supplement 1A). This may be because the peak of the IPSP is often mediated by the GABAB component, in which case it will not be determined by the Cl^-^ reversal potential (e.g. see Figure 6).